# PROVABLE REWARD-AGNOSTIC PREFERENCE-BASED REINFORCEMENT LEARNING

**Wenhao Zhan**
Princeton University
`wenhao.zhan@princeton.edu`

**Masatoshi Uehara**[*]
Genentech
`uehara.masatoshi@gene.com`

**Wen Sun**
Cornell University
`ws455@cornell.edu`

**Jason D. Lee**
Princeton University
`jasonlee@princeton.edu`

## ABSTRACT

Preference-based Reinforcement Learning (PbRL) is a paradigm in which an RL agent learns to optimize a task using pair-wise preference-based feedback over trajectories, rather than explicit reward signals. While PbRL has demonstrated practical success in fine-tuning language models, existing theoretical work focuses on regret minimization and fails to capture most of the practical frameworks. In this study, we fill in such a gap between theoretical PbRL and practical algorithms by proposing a theoretical reward-agnostic PbRL framework where exploratory trajectories that enable accurate learning of hidden reward functions are acquired before collecting any human feedback. Theoretical analysis demonstrates that our algorithm requires less human feedback for learning the optimal policy under preference-based models with linear parameterization and unknown transitions, compared to the existing theoretical literature. Specifically, our framework can incorporate linear and low-rank MDPs with efficient sample complexity. Additionally, we investigate reward-agnostic RL with action-based comparison feedback and introduce an efficient querying algorithm tailored to this scenario.

## 1 INTRODUCTION

Reinforcement learning algorithms train agents to optimize rewards of interests. However, setting an appropriate numerical reward can be challenging in practical applications (e.g., design a reward function for a robot arm to learn to play table tennis), and optimizing hand-crafted reward functions can lead to undesirable behavior when the reward function does not align with human intention. To overcome this challenge, there has been a recent surge of interest in Preference-based Reinforcement Learning (PbRL) with human feedback. In PbRL, the agent does not receive a numerical reward signal, but rather receives feedback from a human expert in the form of preferences, indicating which state-action trajectory is preferred in a given pair of trajectories. PbRL has gained considerable attention in various domains, including NLP (Ziegler et al., 2019; Stiennon et al., 2020; Wu et al., 2021; Nakano et al., 2021; Ouyang et al., 2022; Glaese et al., 2022; Ramamurthy et al., 2022; Liu et al., 2023), robot learning (Christiano et al., 2017; Brown et al., 2019; Shin et al., 2023), and recommender systems (Xue et al., 2022).

Despite the promising applications of PbRL in various areas, there are only a few provably efficient algorithms (also known as PAC RL) for this purpose (Pacchiano et al., 2021; Chen et al., 2022b). These algorithms focus on regret minimization and iterate through the following processes: collecting new trajectories from the environment, obtaining human feedback on the trajectories, and learning hidden reward functions as well as the dynamic model from the human feedback. However, this approach can be slow and expensive in practice as it requires humans in every iteration of dynamic model selection process, which is not as easy as it may sound. For example, interactive decision-making algorithms that put human in the loop of model learning process such as DAgger (Ross et al., 2011) can become impractical when applied to some real-world robotics applications, as has been observed in prior works (Ross et al., 2013; Laskey et al., 2016). In contrast, in practical PbRL

---

[*]This work was done at Cornell University.

applications like InstructGPT (Ouyang et al., 2022) and PEBBLE (Lee et al., 2021), the majority of preference data are collected by crowdsourcing prompts from the entire world and the supervised or heuristic policies, therefore most of the human labeling process does not depend on the training steps afterward. Another line of work (Zhu et al., 2023) focuses on purely offline RL algorithms to learn a near-optimal policy from offline trajectories with good coverage (e.g., offline data that covers some high-quality policies' traces). Nevertheless, it is unclear how to obtain such high-quality offline data a priori (Chen and Jiang, 2019).

To fill in such a gap between theoretical work and practical applications in PbRL, we propose a new theoretical method that lies in between purely online and purely offline methods for PbRL, resembling the framework of InstructGPT and PEBBLE. Our algorithm first collects state-action trajectories from the environment without human feedback. In this step, we design a novel sampling procedure to acquire exploratory trajectories that facilitate the subsequent learning of reward functions which is fully reward-agnostic. In the second step, we collect preference feedback on the collected trajectories from human experts. In the third step, we aim to learn the underlying hidden reward functions using the collected trajectories in the first step and preference feedback in the second step. In the fourth step, we learn the optimal policy by solving the offline RL problem under the learned reward function. Our approach can be understood as performing experimental design for PbRL, which allows us to separate the data-collection process from the process of querying human feedback, eliminating the need for constantly keeping human in the training loop. For instance, we only need to keep human experts in step 2 above, while we can freely perform hyperparameter tuning / model selection for the rest steps without requiring human experts sitting next to the computers. This process can significantly reduce the burden from human experts.

Our contributions can be summarized as follows:

- We propose an efficient experimental design algorithm for PbRL. Our algorithm is specifically designed for linear reward parametrization, which is commonly used in models such as the Bradley-Terry-Luce model, and can handle unknown transitions. This flexibility allows us to handle non-tabular transition models like low-rank MDPs (Agarwal et al., 2020a) and linear MDPs (Jin et al., 2019). To the best of the our knowledge, existing works with statistical guarantees cannot incorporate these models efficiently. Notably, our experimental design algorithm does not depend on any information of the reward and is reward-agnostic. Therefore, the collected trajectories can indeed be reused for learning many reward functions at the same time.

- Our key idea is to decouple the interaction with the environment and the collection of human feedback. This decoupling not only simplifies the process of obtaining human feedback in practice but also results in a significant reduction in the sample complexity associated with human feedback compared to existing works (Pacchiano et al., 2021; Chen et al., 2022b). This improvement is particularly valuable as collecting human feedback is often a resource-intensive process.

- To circumvent the scaling with the maximum per-trajectory reward in the trajectory-based comparison setting, we further investigate preference-based RL with action-based comparison and propose a provably efficient algorithm for this setting. We show that in this case the sample complexity only scales with the bound of the advantage functions of the optimal policy, which can be much smaller than the maximum per-trajectory reward (Ross et al., 2011; Agarwal et al., 2019).

## 1.1 RELATED WORKS

We refer the readers to Wirth et al. (2017) for an overview of Preference-based RL (PbRL). PbRL has been well-explored in bandit setting under the notion of dueling bandits (Yue et al., 2012; Zoghi et al., 2014; Dudík et al., 2015), where the goal is to find the optimal arm in the bandit given human preference over action pairs. For MDPs, in addition to Pacchiano et al. (2021); Chen et al. (2022b), which we compare in the introduction, Novoseller et al. (2020); Xu et al. (2020) have also developed algorithms with sample complexity guarantees. Novoseller et al. (2020) proposes a double posterior sampling algorithm with an asymptotic regret sublinear in the horizon $H$. Xu et al. (2020) proposes a PAC RL algorithm but relies on potentially strong assumptions such as Strong Stochastic Transitivity. Note both of Novoseller et al. (2020); Xu et al. (2020) are limited to the tabular setting.

Our algorithm shares a similar concept with reward-free RL which focuses on exploration in the state-action space without using explicit rewards. Reward-free RL has been studied in many MDPs such as tabular MDPs (Jin et al., 2020a), linear MDPs (Wang et al., 2020), low-rank MDPs (Agarwal

et al., 2020a) and several other models (Chen et al., 2022a; Zanette et al., 2020; Qiu et al., 2021). The goal of reward-free RL is to gather exploratory state-action data to address the challenge of unknown *transitions* before *observing rewards*. In contrast, our approach aims to design a single exploration distribution from which we can draw trajectory pairs to solicit human feedback for learning reward functions. Our setting can be considered as an experimental design for PbRL.

## 2 PRELIMINARIES

We introduce our formulation of Markov decision processes (MDPs) and PbRL.

### 2.1 MDPS WITH LINEAR REWARD PARAMETRIZATION

We consider a finite-horizon MDP $\mathcal{M} = (\mathcal{S}, \mathcal{A}, P^*, r^*, H)$, where $\mathcal{S}$ is the state space, $\mathcal{A}$ is the action space, $P^* = \{P_h^*\}_{h=1}^H$ is the ground-truth transition dynamics, $r^* = \{r_h^*\}_{h=1}^H$ is the ground-truth reward function, and $H$ is the horizon. Specifically, for each $h \in [H]$ ($[H] := (1, \cdots, H)$), $P_h^* : \mathcal{S} \times \mathcal{A} \to \Delta(\mathcal{S})$ and $r_h^* : \mathcal{S} \times \mathcal{A} \to [0, 1]$ represent the transition and reward function at step $h$, respectively. Moreover, we use $P_1(\cdot)$ to denote the initial state distribution. Here, both $r^*, P^*$ are unknown to the learner. In this work, we assume that the cumulative reward of any trajectory $\tau = (s_h, a_h)_{h=1}^H$ does not exceed $r_{\max}$, i.e., $\sum_{h=1}^H r_h(s_h, a_h) \leq r_{\max}$.

**Policies and value functions.** A policy $\pi = \{\pi_h\}_{h=1}^H$ where $\pi_h : \mathcal{S} \to \Delta(\mathcal{A})$ for each $h \in [H]$ characterizes the action selection probability for every state at each step. In this paper, we assume the policy belongs to a policy class $\Pi$, which can be infinite. Given a reward function $r$ and policy $\pi$, the associated value function and Q function at time step $h$ are defined as follows: $V_h^{r,\pi}(s) = \mathbb{E}_{\pi,P^*}[\sum_{h'=h}^H r_h(s_h, a_h)|s_h = s], Q_h^{r,\pi}(s, a) = \mathbb{E}_{\pi,P^*}[\sum_{h'=h}^H r_h(s_h, a_h)|s_h = s, a_h = a]$. Here, $\mathbb{E}_{\pi,P^*}[\cdot]$ represents the expectation of the distribution of the trajectory induced by a policy $\pi$ and the transition $P^*$. We use $V^{r,\pi}$ to denote the expected cumulative rewards of policy $\pi$ with respect to reward function $r$ under $P^*$, i.e., $V^{r,\pi} := \mathbb{E}_{s\sim P_1^*} V_1^{r,\pi}(s)$, and use $V^{r,*}$ to denote the maximal expected cumulative rewards with respect to reward function $r$ under $P^*$, i.e., $V^{r,*} := \max_{\pi \in \Pi} V^{r,\pi}$. In particular, let $\pi^*$ denote the best policy in $\Pi$ with respect to $r^*$, i.e., $\arg\max_{\pi \in \Pi} V^{r^*,\pi}$. In contrast, we denote the globally optimal policy by $\pi_{\mathrm{g}} := \arg\max_{\pi \in \Pi_{\mathrm{Mar}}} V^{r^*,\pi}$ where $\Pi_{\mathrm{Mar}}$ is the set of all Markovian policies. Note that when $\Pi \neq \Pi_{\mathrm{Mar}}$, $\pi^*$ might not be optimal compared to $\pi_{\mathrm{g}}$.

**Linear reward parametrization.** To learn the unknown reward function, it is necessary to make structural assumptions about the reward. We consider a setting where the true reward function possesses a linear structure:

**Assumption 1** (Linear Reward Parametrization). *We assume MDP has a linear reward parametrization with respect to (w.r.t.) known feature vectors $\phi_h(s, a) \in \mathbb{R}^d$. Specifically, for each $h \in [H]$, there exists an unknown vector $\theta_h^* \in \mathbb{R}^d$ such that $r_h^*(s, a) = \phi_h(s, a)^\top \theta_h^*$ for all $(s, a) \in \mathcal{S} \times \mathcal{A}$. For technical purposes, we suppose for all $s \in \mathcal{S}, a \in \mathcal{A}, h \in [H]$, we have $\|\phi_h(s, a)\| \leq R, \|\theta_h^*\| \leq B$.*

Note when $d = |\mathcal{S}||\mathcal{A}|$ and setting $\phi_h(s, a)$ as one-hot encoding vectors, we can encompass the tabular setting. Linear reward parametrization is commonly used in the literature of preference-based RL with statistical guarantees (Pacchiano et al., 2021; Zhu et al., 2023). See Appendix A for more details.

**Notation.** We use $r^*(\tau) := \sum_{h=1}^H r_h^*(s_h, a_h)$ to denote the ground-truth cumulative rewards of trajectory $\tau$. In particular, $r^*(\tau) = \langle \phi(\tau), \theta^* \rangle$ where $\phi(\tau) := [\phi_1(s_1, a_1)^\top, \cdots, \phi_H(s_H, a_H)^\top]^\top, \theta^* := [\theta_1^{*\top}, \cdots, \theta_H^{*\top}]^\top$. We use $\phi(\pi)$ to denote $\mathbb{E}_{\tau\sim(\pi,P^*)}[\phi(\tau)]$ for simplicity. We also use $\Theta(B)$ to denote the set $\{\theta \in \mathbb{R}^d : \|\theta\| \leq B\}$ and $\Theta(B, H)$ to denote the set $\{\theta \in \mathbb{R}^{Hd} : \theta = [\theta_1^\top, \cdots, \theta_H^\top]^\top, \theta_h \in \Theta(B), \forall h \in [H]\} \cap \{\theta \in \mathbb{R}^{Hd} : \langle \phi(\tau), \theta^* \rangle \leq r_{\max}, \forall \tau\}$. We use the notation $f = O(g)$ when there exists a universal constant $C > 0$ such that $f \leq Cg$ and $\widetilde{O}(g) := O(g \log g)$.

### 2.2 PREFERENCE-BASED REINFORCEMENT LEARNING

In this paper, we consider a framework for PbRL that mainly consists of the following four steps:

- **Step 1**: Collect a dataset of trajectory pairs $\mathcal{D}_{\mathrm{reward}} = (\tau^{n,0}, \tau^{n,1})_{n=1}^N$ in a reward-agnostic fashion, where $\tau^{n,i} = \{s_h^{n,i}, a_h^{n,i}, s_{h+1}^{n,i}\}_{h=1}^H$ for $n \in [N]$ and $i \in (0, 1)$.

- **Step 2**: Obtain preference feedback from human experts for each pair of trajectories in $\mathcal{D}_{\mathrm{reward}}$. Namely, if trajectory $\tau^{n,1}$ is preferred over $\tau^{n,0}$, then assign $o^n = 1$, otherwise assign $o^n = 0$.

---

**Algorithm 1 REGIME**: Experimental Design for Querying Human Preference

---

1: **Input**: Regularization parameter $\lambda$, model estimation accuracy $\epsilon'$, parameters $\epsilon, \delta$.
2: Initialize $\Sigma_1 = \lambda I$
3: Estimate model $\widehat{P} \leftarrow \mathcal{P}(\Pi, \epsilon', \delta/4)$ (Possibly, requires the interaction with the **enviroment**.)
4: **for** $n = 1, \cdots, N$ **do**
5:     Compute $(\pi^{n,0}, \pi^{n,1}) \leftarrow \arg\max_{\pi^0, \pi^1 \in \Pi} \|\widehat{\phi}(\pi^0) - \widehat{\phi}(\pi^1)\|_{\widehat{\Sigma}_n^{-1}}$.
6:     Update $\widehat{\Sigma}_{n+1} = \widehat{\Sigma}_n + (\widehat{\phi}(\pi^0) - \widehat{\phi}(\pi^1))(\widehat{\phi}(\pi^0) - \widehat{\phi}(\pi^1))^\top$.
7: **end for**
8: **for** $n = 1, \cdots, N$ **do**
9:     Collect a pair of trajectories $\tau^{n,0}, \tau^{n,1}$ from the **enviroment** by $\pi^{n,0}, \pi^{n,1}$, respectively.
10:     Add it to $\mathcal{D}_{\text{reward}}$.
11: **end for**.
12: Obtain the preference labels $\{o^n\}_{n=1}^N$ for $\mathcal{D}_{\text{reward}}$ from **human experts**.
13: Run MLE $\widehat{\theta} \leftarrow \arg\max_{\theta \in \Theta(B,H)} L(\theta, \mathcal{D}_{\text{reward}}, \{o^n\}_{n=1}^N)$ where $L(\theta, \mathcal{D}_{\text{reward}}, \{o^n\}_{n=1}^N)$ is defined in (1).
14: Return $\widehat{\pi} = \arg\max_{\pi \in \Pi}\langle \widehat{\phi}(\pi), \widehat{\theta}\rangle$.

---

- **Step 3**: Estimate the ground truth reward using the dataset $\mathcal{D}_{\text{reward}}$ and preference labels $\{o^n\}_{n=1}^N$.

- **Step 4**: Run RL algorithms (either online or offline) using the learned rewards and obtain a policy $\widehat{\pi}$ that maximizes the cumulative learned rewards.

The above framework has been applied in practical applications, such as PEBBLE (Lee et al., 2021). However, these algorithms lack provable sample efficiency guarantees. In particular, it remains unclear in the literature how to collect the trajectories in **Step 1** to enable accurate estimation of the ground truth reward. In our work, we strive to develop a concrete algorithm that adheres to the above framework while ensuring theoretical sample efficiency. We also emphasize that step 1 is reward-agnostic, and the collected dataset can be re-used for learning many different rewards as long as they are linear in the feature $\phi$.

**Preference model.** In this work, we assume the preference label follows the Bradley-Terry-Luce (BTL) model (Bradley and Terry, 1952) in Step 2, i.e., we have the following assumption:

**Assumption 2.** *Suppose for any pair of trajectory $(\tau^0, \tau^1)$, we have*

$$\mathbb{P}(o = 1) = \mathbb{P}(\tau^1 \succ \tau^0) = \sigma(r^*(\tau^1) - r^*(\tau^0)) = \frac{\exp(r^*(\tau^1))}{\exp(r^*(\tau^0)) + \exp(r^*(\tau^1))},$$

*where $o$ is the human preference over $(\tau^0, \tau^1)$ and $\sigma(\cdot)$ is the sigmoid function.*

Our analysis will leverage the quantity $\kappa := \sup_{|x| \leq r_{\max}} |1/\sigma'(x)| = 2 + \exp(2r_{\max}) + \exp(-2r_{\max})$ to measure the difficulty of estimating the true reward from the BTL preference model.

## 3 ALGORITHM: REGIME

We propose an algorithm specifically designed for the PbRL setting when the transitions are unknown. In order to handle unknown transitions, we use the following mild oracle:

**Definition 1** (Reward-free RL oracle). *A reward-free learning oracle $\mathcal{P}(\Pi, \epsilon, \delta)$ can return an estimated model $\widehat{P}$ such that with probability at least $1 - \delta$, we have for all policy $\pi \in \Pi$ and $h \in [H], s \in \mathcal{S}, a \in \mathcal{A}, \|\widehat{P}_1(\cdot) - P_1^*(\cdot)\|_1 \leq \epsilon', \mathbb{E}_{\pi,P^*}[\|\widehat{P}_h(\cdot|s,a) - P_h^*(\cdot|s,a)\|_1] \leq \epsilon'$ where $\|\cdot\|_1$ denotes total variation distance (i.e., $\ell_1$-norm).*

This oracle necessitates accurate model learning through interactions with the environment. The required guarantee is relatively mild since we do not require a point-wise error guarantee, but rather an expectation-based guarantee under the ground truth transition. This oracle holds true not only in tabular MDPs (Jin et al., 2020a), but also in low-rank MDPs (Agarwal et al., 2020a; 2022), where the only assumption is the low-rank property of the transition dynamics, and features could be *unknown* to the learner. Low-rank MDPs find wide application in practical scenarios, including blocked MDPs (Du et al., 2019; Zhang et al., 2020a;b; Sodhani et al., 2021; 2022).

### 3.1 ALGORITHM

The algorithm is described in Algorithm 1. Given a learned model $\hat{P}$, we use $\hat{\phi}(\pi) = \mathbb{E}_{\tau \sim (\pi, \hat{P})}[\phi(\tau)]$ to estimate $\phi(\pi) := \mathbb{E}_{\tau \sim (\pi, P^\star)}[\phi(\tau)]$. The algorithm mainly consists of four steps as follows.

**Step 1: Collection of state-action trajectories by interacting with the environment (Line 4–11).** To learn the ground truth reward function, we collect exploratory state-action trajectories that cover the space spanned by $\phi(\cdot)$ before collecting any human feedback. To achieve this, at each iteration, we identify a set of explorative policy pairs that are not covered by existing data. We measure the extent to which the trajectory generated by $(\pi_0, \pi_1)$ can be covered by computing the norm of $\widehat{\phi}(\pi_0) - \widehat{\phi}(\pi_1)$ on the metric induced by the inverse covariance matrix $\Sigma_n^{-1}$ at time step $n$. After iterating this procedure $N$ times and obtaining sets of policies $\{(\pi^{n,0}, \pi^{n,1})\}_{n=1}^N$, we sample $N$ exploratory trajectory pairs by executing the policy pairs $(\pi^{n,0}, \pi^{n,1})$ for $n \in [N]$. Notably, this trajectory-collection process is reward-agnostic and thus the collected samples can be used to learn multiple rewards in multi-task RL.

**Step 2: Collection of preference feedback by interacting with human experts (Line 12).** If trajectory $\tau^{n,1}$ is preferred over $\tau^{n,0}$, then assign $o^n = 1$, otherwise assign $o^n = 0$.

**Step 3: Reward learning via MLE (Line 13).** We adopt the widely-used maximum likelihood estimation (MLE) approach to learn the reward function, which has also been employed in other works Ouyang et al. (2022); Christiano et al. (2017); Brown et al. (2019); Shin et al. (2023); Zhu et al. (2023). Specifically, we learn the reward model by maximizing the log-likelihood $L(\theta, \mathcal{D}_{\text{reward}}, \{o^n\}_{n=1}^N)$:

$$\sum_{n=1}^N \log\left(o^n \cdot \sigma(\langle \theta, \phi(\tau^{n,1}) - \phi(\tau^{n,0})\rangle) + (1 - o^n) \cdot \sigma(\langle \theta, \phi(\tau^{n,0}) - \phi(\tau^{n,1})\rangle)\right). \quad (1)$$

**Step 4: RL with respect to learned rewards (Line 14).** We obtain the near-optimal policy that maximizes the cumulative learned rewards.

Our algorithm differs significantly from the algorithms proposed in Pacchiano et al. (2021); Chen et al. (2022b). In their algorithms, they repeat the following steps: (a) collect new trajectories from the environment using policies based on the current learned reward and transition models, (b) collect human feedback for the obtained trajectories, (c) update the reward and transition models. A potential issue with this approach is that every time human feedback is collected, agents need to interact with the environment, causing a wait time for humans. In contrast, our algorithm first collects exploratory trajectories without collecting any human feedback in Step 1. Then, we query human feedback and learn the reward model in Step 2-3. As a result, we decouple the step of collecting exploratory data from that of collecting human feedback. Hence, in our algorithm, we can efficiently query human feedback in parallel, mirroring common practice done in InstructGPT. Moreover, our algorithm's design leads to lower sample complexity for both trajectory pairs and human feedback than Pacchiano et al. (2021); Chen et al. (2022b). See Appendix A for our technical novelty.

**Remark 1.** *Our collection method in Step 1 shares a similar idea to active learning. See Appendix A.*

**Remark 2.** *The majority of computational cost lies in line 5 in Algorithm 1. To implement the algorithm, gradient ascent can be applied here to solve the optimization problem. See Appendix A.*

**Remark 3.** *In Step 4 (Line 14), it is not necessary to use the same $\hat{P}$ as in Line 3. Instead, any sample-efficient RL algorithm can be employed w.r.t. the learned reward such as Lee et al. (2021).*

### 3.2 ANALYSIS

Now we provide the sample complexity of Algorithm 1 as shown in the following theorem.

**Theorem 1.** *Let*

$$\lambda \geq 4HR^2, \quad N \geq \widetilde{\mathcal{O}}\left(\frac{\lambda \kappa^2 B^2 R^2 H^4 d^2 \log(1/\delta)}{\epsilon^2}\right), \quad \epsilon' \leq \frac{\epsilon}{6BR\sqrt{H^5 d \log N}},$$

*Then under Assumption 1 and 2, with probability at least $1 - \delta$, we have*

$$V^{r^*, \widehat{\pi}} \geq V^{r^*, *} - \epsilon.$$

*Note the sample complexity in Theorem 1 does not depend on the complexity of $\Pi$ and thus we can learn arbitrary policy classes. When $\Pi = \Pi_{\text{Mar}}$, we have $\pi^* = \pi_g$ and thus we can compete against the global optimal policy.*

Since the sample complexity of human feedback, denoted by $N_{\text{hum}}$, is equal to $N$, Theorem 1 shows that the sample complexity of human feedback required to learn an $\epsilon$-optimal policy scales with $\tilde{O}(1/\epsilon^2)$ and is polynomial in the norm bounds $B, R$, the horizon $H$, and the dimension of the feature space $d$. Notably, the sample complexity of human feedback $N_{\text{hum}}$ only depends on the structural complexity of the reward function, regardless of the underlying transition model. This is because while our theorem requires that the learned transition model is accurate enough ($\epsilon' \leq \frac{\epsilon}{6BRH^2}$), we do not need human feedback to learn the transition model for this purpose. This property of our algorithm is particularly desirable when collecting human feedback is much more expensive than collecting trajectories from the environment. Existing works with sample-efficient guarantees, such as Pacchiano et al. (2021); Chen et al. (2022b), do not have this property. Our algorithm's favorable property can be attributed to the careful design of the algorithm, where the step of collecting trajectories and learning transitions is reward-agnostic and thus separated from the step of collecting human feedback and learning rewards. Furthermore, note that our results algorithm indeed works beyond low-rank MDPs, as long as there exists a suitable reward-free model-learning oracle. See Appendix A for more details.

As the most relevant work, we compare our results with Pacchiano et al. (2021), which considers online learning in PbRL with unknown tabular transition models and linear reward parameterization. Let $N_{\text{tra}}$ and $N_{\text{hum}}$ denote the number of required trajectory pairs and human feedback, respectively. Then, to obtain an $\epsilon$-optimal policy, the algorithm in Pacchiano et al. (2021, Theorem 2) requires:

$$N_{\text{tra}} = N_{\text{hum}} = \widetilde{\mathcal{O}}\left( \frac{|\mathcal{S}|^2|\mathcal{A}|d + \kappa^2 d^2}{\epsilon^2} \log \frac{1}{\delta} \right).$$

Here we omit the dependence on $B, R, H$ to facilitates the comparison. In contrast, in the setting considered in Pacchiano et al. (2021), by leveraging the reward-free learning oracle from Jin et al. (2020a), our algorithm achieves the following sample complexity:

$$N_{\text{tra}}\widetilde{\mathcal{O}}\left( \frac{|\mathcal{S}|^2|\mathcal{A}|d + \kappa^2 d^2}{\epsilon^2} \log \frac{1}{\delta} \right), N_{\text{hum}} = \widetilde{\mathcal{O}}\left( \frac{\kappa^2 d^2}{\epsilon^2} \log \frac{1}{\delta} \right),$$

where the number of required trajectory-pairs comes from Jin et al. (2020a)[Lemma 3.6]. We observe that our algorithm achieves a better sample complexity for human feedbacks than the previous work while retaining the total trajectory complexity. In particular, our algorithm has the advantage that $N_{\text{hum}}$ depends only on the feature dimension $d$ and not on $|\mathcal{S}|$ or $|\mathcal{A}|$. This improvement is significant since obtaining human feedback is often costly. Lastly, we note that a similar comparison can be made to the work of Chen et al. (2022b), which considers reward and transition models with bounded Eluder dimension.

## 4 REGIME IN LINEAR MDPS

So far, we have considered PbRL given reward-free RL oracle satisfying Definition 1. Existing works have shown the existence of such a model-based reward-free RL oracle in low-rank MDPs (Agarwal et al., 2020a; 2022). However, these results have not been extended to linear MDPs (Jin et al., 2020b) where model-free techiniques are necessary. Linear MDPs are relevant to our setting because linear reward parametrization naturally holds in linear MDPs. Unfortunately, a direct reduction from linear MDPs to low-rank MDPs may introduce a dependence on the cardinality of $\mathcal{S}$ without assuming strong inductive bias in the function class. In this section, we propose a model-free algorithm that can overcome this dependence by making slight modifications to Algorithm 1. We begin by providing the definition of linear MDPs.

**Assumption 3** (Linear MDPs (Jin et al., 2020b)). *We suppose MDP is linear with respect to some known feature vectors $\phi_h(s, a) \in \mathbb{R}^d (h \in [H], s \in \mathcal{S}, a \in \mathcal{A})$. More specifically, if for each $h \in [H]$, there exist $d$ unknown signed measures $\mu_h^* = (\psi_h^{(1)}, \cdots, \psi_h^{(d)})$ over $\mathcal{S}$ and an unknown vector $\theta_h^* \in \mathbb{R}^d$ such that $P_h^*(\cdot|s, a) = \phi_h(s, a)^\top \mu_h^*(\cdot)$ and $r_h^*(s, a) = \phi_h(s, a)^\top \theta_h^*$ for all $(s, a) \in \mathcal{S} \times \mathcal{A}$. For technical purposes, we suppose the norm bound $\|\mu_h^*(s)\|_2 \leq \sqrt{d}$ for any $s \in \mathcal{S}$.*

In addition, we use $\mathcal{N}_\Pi(\epsilon)$ to denote the covering number of $\Pi$, which is defined as follows:

**Definition 2** ($\epsilon$-covering number). *The $\epsilon$-covering number of the policy class $\Pi$, denoted by $\mathcal{N}_\Pi(\epsilon)$, is the minimum integer $n$ such that there exists a subset $\Pi' \subset \Pi$ with $|\Pi'| = n$ and for any $\pi \in \Pi$ there exists $\pi' \in \Pi'$ such that $\max_{s \in \mathcal{S}, h \in [H]} \|\pi_h(\cdot|s) - \pi'_h(\cdot|s)\|_1 \leq \epsilon$.*

---

**Algorithm 2 REGIME-lin**

---

**Input**: Regularization parameter $\lambda$, feature estimation sample complexity $K$.

Call Algorithm 4 with generating $K$ trajectories by interacting with the **environment**.

Call Algorithm 5 with reward function $(r_{h'}^{h,j})_{h' \in [H]}$ to estimate $(\widehat{\phi}(\pi))_{h,j}$ for all $\pi \in \Pi, h \in [H], j \in [d]$ using $K$ trajectories. Let $\widehat{\phi}(\pi) = [\widehat{\phi}_1(\pi), \cdots, \widehat{\phi}_H(\pi)]$ where the $j$-th entry of $\widehat{\phi}_h(\pi)$ is $(\widehat{\phi}(\pi))_{h,j}$.

**for** $n = 1, \cdots, N$ **do**

    Compute $(\pi^{n,0}, \pi^{n,1}) \leftarrow \arg\max_{\pi^0, \pi^1 \in \Pi} \|\widehat{\phi}(\pi^0) - \widehat{\phi}(\pi^1)\|_{\widehat{\Sigma}_n^{-1}}$.

    Update $\widehat{\Sigma}_{n+1} = \widehat{\Sigma}_n + (\widehat{\phi}(\pi^0) - \widehat{\phi}(\pi^1))(\widehat{\phi}(\pi^0) - \widehat{\phi}(\pi^1))^\top$.

**end for**

**for** $n = 1, \cdots, N$ **do**

    Collect a pair of trajectories $\tau^{n,0}, \tau^{n,1}$ from the **environment** by $\pi^{n,0}, \pi^{n,1}$, respectively.

    Add $(\tau^{n,0}, \tau^{n,1})$ to $\mathcal{D}_{\mathrm{reward}}$.

**end for**

Obtain the preference labels $\{o^{(n)}\}_{n=1}^N$ from **human experts**.

Run MLE $\widehat{\theta} \leftarrow \arg\min_{\theta \in \Theta(B,H)} L_\lambda(\theta, \mathcal{D}_{\mathrm{reward}}, \{o^{(n)}\}_{n=1}^N)$.

Return $\widehat{\pi} = \arg\max_{\pi \in \Pi} \widehat{V}^\pi(\widehat{r})$ where $\widehat{V}^\pi(\widehat{r})$ is obtained by calling Algorithm 5 with reward function $\widehat{r} = \{\widehat{r}_h\}_{h=1}^H$ for all $\pi$ where $\widehat{r}_h(s,a) = \langle \phi_h(s,a), \widehat{\theta} \rangle$.

---

## 4.1 ALGORITHM

The reward-free RL oracle that satisfies Definition 1 for learning accurate transitions may be excessively strong for linear MDPs. Upon closer examination of Algorithm 1, it becomes apparent that the learned transition model is solely used for estimating $\phi(\pi)$. Therefore, our approach focuses on achieving a precise estimation of $\phi(\pi)$.

Our main algorithm is described in Algorithm 2 with subroutines for estimating $\widehat{\phi}(\pi)$. The overall structure of the primary algorithm resembles that of Algorithm 1. The key distinction lies in the part to accurately estimate $\widehat{\phi}(\pi)$ within the subroutines, without relying on the abstract reward-free RL oracle (Definition 1). In the following, we provide a brief explanation of these subroutines. The detailed descriptions of these subroutines is deferred to Algorithm 4 and 5 in Appendix B.

**Collecting exploratory data to learn transitions.** Being inspired by the approach in Jin et al. (2020b); Wang et al. (2020), we construct an exploratory dataset by running LSVI-UCB (Jin et al., 2020b) with rewards equivalent to the bonus. Specifically, in the $k$-th iteration, we recursively apply the least square value iteration with a bonus term $\{b_h^k(s,a)\}_{h=1}^H$, which is introduced to induce exploration. This process yields an exploratory policy $\pi^k$ based on exploratory rewards $\{r_h^k\}_{h=1}^H$, where $r_h^k = b_h^k/H$. We then collect a trajectory by executing policy $\pi^k$. By repeating this procedure for $K$ iterations, we accumulate an exploratory dataset. The detailed algorithm is provided in Appendix B (Algorithm 4). It is important to note that this step involves generating $K$ trajectories through interactions with the environment.

**Estimating $\phi(\pi)$ using the exploratory data.** Let $(\phi(\pi))_{h,j}$ denote the $j$-th entry of $\phi_h(\pi) := \mathbb{E}_\pi[\phi_h(s_h, a_h)]$. Then to estimate $\phi(\pi)$, we only need to estimate $(\phi(\pi))_{h,j}$ for all $h \in [H], j \in [d]$. Note that for all $\pi \in \Pi$, we have $\phi(\pi) = \left[ \mathbb{E}_{\pi, P^*}[\phi_1(s_1, a_1)^\top], \cdots, \mathbb{E}_{\pi, P^*}[\phi_H(s_H, a_H)^\top] \right]^\top$. Here, the key observation is that $(\phi(\pi))_{h,j}$ is exactly the expected cumulative rewards with respect to the following reward function $r_{h'}^{h,j}(s,a) = \phi_{h'}(s,a)^\top \theta_{h'}^{h,j}$ for all $h' \in [H]$ (up to an $R$ factor) where $\theta_{h'}^{h,j} = \frac{1}{R} \cdot e_j$ for $h' = h$ and $\theta_{h'}^{h,j} = 0$, otherwise ($h' \neq h$). Here $e_j$ is the one-hot encoding vector whose $j$-th entry is 1. Therefore, with the collected dataset, we can run the least square policy evaluation to estimate $(\phi(\pi))_{h,j}$. The detail is in Algorithm 5 in Appendix B.

## 4.2 ANALYSIS

Now we present the sample complexity of Algorithm 2. The formal statement and proof are deferred to Appendix B and E.1.

**Theorem 2** (Informal). *By choosing parameters in an appropriate way and setting*

$$K \geq \widetilde{\mathcal{O}}\left(\frac{H^9 B^2 R^4 d^5 \log(\mathcal{N}_\Pi(\epsilon')/\delta)}{\epsilon^2}\right), \quad N \geq \widetilde{\mathcal{O}}\left(\frac{\lambda \kappa^2 B^2 R^2 H^4 d^2 \log(1/\delta)}{\epsilon^2}\right), \epsilon' = \frac{\epsilon}{72 B R^2 H \sqrt{d^H K^{H-1}}}$$

*under Assumption 1,2, and 3, with probability at least $1-\delta$, we have $V^{r^*,\hat{\pi}} \geq V^{r^*,*} - \epsilon$. Furthermore, by selecting a policy class $\Pi$ properly, we have $V^{r^*,\hat{\pi}} \geq V^{r^*,\pi_g} - 2\epsilon$ by replacing $\log(\mathcal{N}_\Pi(\epsilon')/\delta) = Hd \log\left(\frac{12WR}{\epsilon'}\right)$ where $W = \frac{\left(B + (H+\epsilon)\sqrt{d}\right) H \log|\mathcal{A}|}{\epsilon}$.*

The first statement says Algorithm 2 can learn an $\epsilon$-optimal policy with the number of trajectory-pairs and human feedbacks as follows:

$$N_{\text{tra}} = K + N = \widetilde{\mathcal{O}}\left(\frac{d^5 \log \mathcal{N}_\Pi(\epsilon') + \kappa^2 d^2}{\epsilon^2}\right), N_{\text{hum}} = \widetilde{\mathcal{O}}\left(\frac{\kappa^2 d^2}{\epsilon^2}\right).$$

Since the sample complexity depends on the covering number of $\Pi$, we need to carefully choose the policy class. When we choose $\Pi$ to be the log-linear policy class:

$$\Pi = \left\{\pi = \{\pi_h^\zeta\}_{h=1}^H : \pi_h^\zeta(a|s) = \frac{\exp(\zeta_h^\top \phi_h(s,a))}{\sum_{a' \in \mathcal{A}} \exp(\zeta_h^\top \phi_h(s,a'))}, \zeta_h \in \mathbb{B}(d,W), \forall s \in \mathcal{S}, a \in \mathcal{A}, h \in [H]\right\},$$

although $\pi^* \neq \pi_g$, we can show that the value of $\pi^*$ is close to the value of $\pi_g$ up to $\epsilon$ by setting sufficiently large $W$. This immediately leads to the second statement in Theorem 2. Consequently, to learn an $\epsilon$-global-optimal policy, it is concluded that the number of required trajectory pairs and human feedbacks for Algorithm 2 does not depend on $|\mathcal{S}|$ at all.

Finally, we compare our work to Chen et al. (2022b), as it is the only existing work that addresses provable PbRL with non-tabular transition models. Their algorithm exhibits sample complexities that depend on the Eluder dimension associated with the transition models. However, in linear MDPs, it remains uncertain whether we can get upper-bound on the Eluder dimension without introducing a dependence on $|\mathcal{S}|$. Consequently, our Algorithm 2 is the first provable PbRL algorithm capable of achieving polynomial sample complexity that is independent of $|\mathcal{S}|$ in linear MDPs.

## 5 Regime with Action-Based Comparison

The drawback of the current results is that the sample complexity is dependent on $\kappa$, which can exhibit exponential growth in $r_{\max}$ under the BTL model. This is due to the fact that $\sup_{|x| \leq r_{\max}} |1/\sigma'(x)| = O(\exp(r_{\max}))$. Such dependence on $r_{\max}$ is undesirable, especially when rewards are dense and $r_{\max}$ scales linearly with $H$. Similar limitations are present in existing works, such as Pacchiano et al. (2021); Chen et al. (2022b). To address this challenge, we consider the action-based comparison model (Zhu et al., 2023) in this section. Here, we assume that humans compare two actions based on their optimal Q-values. Given a tuple $(s, a^0, a^1, h)$, the human provides feedback $o$ following

$$\mathbb{P}(o = 1|s, a^0, a^1, h) = \mathbb{P}(a^1 \succ a^0|s,h) = \sigma(A_h^*(s,a^1) - A_h^*(s,a^0)), \tag{2}$$

where $A_h^*$ is the advantage function of the optimal policy. Similar to trajectory-based comparisons with linear reward parametrization, we assume linearly parameterized advantage functions:

**Assumption 4** (Linear Advantage Parametrization). *An MDP has linear advantage functions with respect to some known feature vectors $\phi_h(s,a) \in \mathbb{R}^d (h \in [H], s \in \mathcal{S}, a \in \mathcal{A})$ if for each $h \in [H]$, there exists an unknown vector $\xi_h^* \in \mathbb{R}^d$ such that $A_h^*(s,a) = \phi_h(s,a)^\top \xi_h^*$ for all $(s,a) \in \mathcal{S} \times \mathcal{A}$. We assume for all $s \in \mathcal{S}, a \in \mathcal{A}, h \in [H]$, we have $\|\phi_h(s,a)\| \leq R, \|\xi_h^*\| \leq B$.*

Generally, the value of $|A_h^*(s,a)|$ tends to be much smaller than $H$ since a large value of $|A_h^*(s,a)|$ implies that it may be difficult to recover from a previous incorrect action even under the best policy $\pi^*$ (Ross et al., 2011; Agarwal et al., 2019). Therefore, by defining $B_{\text{adv}} = \sup_{(s,a)} |A_h^*(s,a)|$, we expect that $B_{\text{adv}}$ will be much smaller than $H$, even in scenarios with dense rewards.

In the following discussion, we will use $Z(B,h)$ to denote the convex set $\{\zeta \in \mathbb{R}^d : \|\zeta\| \leq B, \langle \phi_h(s,a), \zeta \rangle \leq B_{\text{adv}}, \forall s \in \mathcal{S}, a \in \mathcal{A}\}$. We consider the setting where $\Pi = \Pi_{\text{Mar}}$ and assume the transition model is known for brevity. In the case of unknown transition models, we can employ the same approach as described in Section 3 with reward-free RL oracles.

We present our algorithm for action-based comparison models in Algorithm 3. In Line 19 we denote

$$L(\xi, \mathcal{D}_{\text{adv}}^h, \{o^{h,n}\}_{n=1}^N) := \sum_{n=1}^N \log\left(o^{h,n} \cdot \sigma(\langle \xi, \phi_h(s^{h,n}, a^{h,n,1}) - \phi(s^{h,n}, a^{h,n,0})\rangle)\right)$$

---

**Algorithm 3 `REGIME-action`**

---

1: **Input**: Regularization parameter $\lambda$.
2: **for** $h = 1, \cdots, H$ **do**
3:     Initialize $\Sigma_{h,1} = \lambda I$.
4:     **for** $n = 1, \cdots, N$ **do**
5:         Compute: $(\pi^{h,n,0}, \pi^{h,n,1}) \leftarrow \arg\max_{\pi^0, \pi^1 \in \Pi} \|\mathbb{E}_{s_h \sim \pi^0}[\phi_h(s_h, \pi^0) - \phi_h(s_h, \pi^1)]\|_{\Sigma_{h,n}^{-1}}$,
6:         where $\phi_h(s, \pi) = \mathbb{E}_{a \sim \pi_h(\cdot|s)}[\phi_h(s, a)]$.
7:         Update:

$$\Sigma_{h,n+1} = \Sigma_{h,n} + (\mathbb{E}_{s_h \sim \pi^{h,n,0}}[\phi_h(s_h, \pi^{h,n,0}) - \phi_h(s_h, \pi^{h,n,1})])$$
$$\cdot (\mathbb{E}_{s_h \sim \pi^{h,n,0}}[\phi_h(s_h, \pi^{h,n,0}) - \phi_h(s_h, \pi^{h,n,1})])^\top$$

8:     **end for**
9: **end for**
10: **for** $h = 1, \cdots, H$ **do**
11:     **for** $n = 1, \cdots, N$ **do**
12:         Sample $s^{h,n}$ at time step $h$ by executing a policy $\pi^{h,n,0} = \{\pi_k^{h,n,0}\}_{k=1}^H$.
13:         Sample actions $a^{h,n,0} \sim \pi_h^{h,n,0}(\cdot|s^{h,n}), a^{h,n,1} \sim \pi_h^{h,n,1}(\cdot|s^{h,n})$.
14:         Add $(s^{h,n}, a^{h,n,0}, a^{h,n,1})$ to $\mathcal{D}_{\text{adv}}^h$.
15:         (These steps involve the interaction with **environment**)
16:     **end for**
17: **end for**
18: Obtain the preference labels $\{o^{h,n}\}_{n=1}^N$ for $\mathcal{D}_{\text{adv}}^h$ from **human experts**.
19: Run MLE $\widehat{\xi}_h \leftarrow \arg\min_{\xi \in Z(B,h)} L(\xi, \mathcal{D}_{\text{adv}}^h, \{o^{h,n}\}_{n=1}^N)$.
20: Compute: for all $s \in \mathcal{S}, a \in \mathcal{A}, h \in [H]$:
21: $\widehat{A}_h(s, a) \leftarrow \phi_h(s, a)^\top \widehat{\xi}_h, \widehat{\pi}_h(s) \leftarrow \arg\max_{a \in \mathcal{A}} \widehat{A}_h(s, a)$.
22: Return $\widehat{\pi} = \{\widehat{\pi}\}_{h=1}^H$.

---

$$+ (1 - o^{h,n}) \cdot \sigma(\langle \xi, \phi_h(s^{h,n}, a^{h,n,0}) - \phi(s^{h,n}, a^{h,n,1})\rangle)\Big),$$

where $\mathcal{D}_{\text{adv}}^h = \{s^{h,n}, a^{h,n,0}, a^{h,n,1}\}_{n=1}^N$.

## 5.1 ANALYSIS

**Theorem 3.** *Let*

$$\lambda \geq 4R^2, \qquad N \geq \widetilde{\mathcal{O}}(\lambda \kappa_{\text{adv}}^2 B^2 R^2 H^2 d^2 \log(1/\delta)/\epsilon^2)$$

*where $\kappa_{\text{adv}} = \sup_{|x| \leq B_{\text{adv}}} |1/\sigma'(x)|$ in* `REGIME-action`. *Then under Assumption 4, with probability at least $1 - \delta$, we have $V^{r^*, \widehat{\pi}} \geq V^{r^*, *} - \epsilon$.*

Theorem 3 demonstrates that for the action-based comparison model, the number of required human feedbacks scales with $\kappa_{\text{adv}}$ instead of $\kappa$. This implies that when $\sigma$ is a commonly used sigmoid function, the sample complexity is exponential in $B_{\text{adv}}$ rather than $r_{\max}$. Crucially, $B_{\text{adv}}$ is always less than or equal to $r_{\max}$, and as mentioned earlier, $B_{\text{adv}}$ can be $o(H)$ even in dense reward settings where $r_{\max} = \Theta(H)$. Consequently, we achieve superior sample complexity compared to the trajectory-based comparison setting.

## 6 SUMMARY

We consider the problem of how to query human feedback efficiently in PbRL, i.e., the experimental design problem in PbRL. In particular, we design a reward-agnostic trajectory collection algorithm for human feedback querying when the transition dynamics is unknown. Our algorithm provably requires less human feedback to learn the true reward and optimal policy than existing literature. Our results also go beyond the tabular cases and cover common MDPs models including linear MDPs and low-rank MDPs. Further, we consider the action-based comparison setting and propose corresponding algorithms to circumvent the exponential scaling with $r_{\max}$ of trajectory-based comparison setting.

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

# A    DISCUSSION

**Validity of Linear Parametrization.**    In this work we consider linear reward parametrization, or more generally, linear trajectory embeddings. Such assumptions are commonly used in the theoretical works of PbRL (Pacchiano et al., 2021; Zhu et al., 2023) and relevant examples can be borrowed from the practical works (Pacchiano et al., 2020; Parker-Holder et al., 2020) like the behavior guided class of algorithms for policy optimization. We admit that our analysis cannot cover all kinds of reward parametrization, but we think it is a reasonable starting point to consider linear function approximation for theoretical analysis.

**Applicability of `REGIME`.**    Our results are not restricted to linear MDPs and low-rank MDPs. Our main result (Theorem 1) only requires a suitable reward-free oracle for dynamics learning and in practice there exist some oracles ready for use, even when the MDPs are very complicated (Xu et al., 2022). This implies that by plugging in these general reward-free oracles, we are able to deal with general MDPs as well.

**Relationship to Active Learning.**    The trajectory collection process in Step 1 of `REGIME` utilizes a similar idea to active learning. In active learning, people choose a trajectory pair to query human feedback which can maximize the information gain in each iteration. Similarly, in our algorithm, the estimated covariance matrix $\widehat{\Sigma}_n$ can be regarded as the current information in $n$-th iteration. Then we choose a pair of policies $(\pi^{n,0}, \pi^{n,1})$ which can maximize the information gain approximately (line 5, it is approximate because we are using an approximate dynamics $\widehat{P}$ for evaluation).

**Implementation of Algorithm 1.**    The majority of computational cost of Algorithm 1 lies in line 5. To implement the algorithm, gradient ascent can be applied here to solve the optimization problem. More specifically, assume that $\pi^0$ and $\pi^1$ are parameterized by $\nu^0$ and $\nu^1$ (for example, they might be parametrized by a neural network). Then the gradient of the objective $f(\nu^0, \nu^1) := \|\widehat{\phi}(\pi^0) - \widehat{\phi}(\pi^1)\|_{\widehat{\Sigma}_n^{-1}}$ with respect to $\nu^0$ (gradient of $\nu^1$ can be similarly computed) can be computed as:

$$\frac{\partial f(\nu^0, \nu^1)}{\partial \nu^0} = 2\frac{\partial \widehat{\phi}(\pi^0)}{\partial \nu^0} \cdot \widehat{\Sigma}_n^{-1} \cdot (\widehat{\phi}(\pi^0) - \widehat{\phi}(\pi^1))$$

Here $\widehat{\phi}(\pi^0)$ and $\widehat{\phi}(\pi^1)$ can be estimated efficiently by simulating $\pi^0$ and $\pi^1$ in $\widehat{P}$. For $\frac{\partial \widehat{\phi}(\pi^0)}{\partial \nu^0}$, we show in Section 4 that each coordinate of $\widehat{\phi}(\pi^0)$ is indeed a value function with respect to a certain reward function under $\widehat{P}$. Therefore we can apply the techniques in policy gradient literature such as REINFORCE to estimate $\frac{\partial \widehat{\phi}(\pi^0)}{\partial \nu^0}$ efficiently.

**Technical Novelty.**    The novelty of our algorithm mainly lies in the reward-agnostic trajectory collection procedure (line 4-11 in Algorithm 1). For the other steps in the framework, we follow the common practice in empirical works and thus they are the same as the practical work. Our reward-agnostic trajectory collection procedure takes inspiration from the existing works in reward-free exploration RL and online PbRL. However, it is not a simple combination of the existing algorithms. More specifically, our analysis has the following challenges that are not solved in the literature:

- First, almost all the reward-free exploration literature is focused on learning the transition dynamics of the MDPs. Our problem is completely different because our main task is to learn the reward function from human preferences. Consequently, our iterative collection procedure (line 4-11 in Algorithm 1) also differs from the reward-free exploration literature. For example, Wang et al. (2020) is the closest work to ours and studies the reward-free exploration in linear MDPs. They utilize a variant of UCB-LSVI, which is quite different from our collection procedure. Correspondingly, we can not directly use the existing analysis either and need to come up with new proofs.

- Second, our framework is also different from the existing online PbRL (Pacchiano et al., 2021; Chen et al., 2022b). In their algorithms, the reward-learning process and exploration phase are not decoupled and the analysis heavily relies on the utilization of optimism. In contrast, our algorithm does not use optimism and thus their analysis techniques also do not apply here.

---

**Algorithm 4 `REGIME-exploration`**

---

**Input**: The number of total episodes $K$, bonus parameter $\beta_{\mathrm{ex}}$ and regularization parameter $\lambda_{\mathrm{ex}}$.

**for** $k = 1, \cdots, K$ **do**

    Initialize: $Q_{H+1}^k(\cdot, \cdot) \leftarrow 0, V_{H+1}^k(\cdot) \leftarrow 0$.

    **for** $h = H, \cdots, 1$ **do**

        Compute the covariance matrix: $\Lambda_h^k \leftarrow \sum_{i=1}^{k-1} \phi_h(s_h^i, a_h^i)\phi_h(s_h^i, a_h^i)^\top + \lambda_{\mathrm{ex}} I$.

        Compute the bonus and reward:

        $b_h^k(\cdot, \cdot) \leftarrow \min\left\{\beta_{\mathrm{ex}}\|\phi_h(\cdot, \cdot)\|_{(\Lambda_h^k)^{-1}}, H - h + 1\right\}$ and $r_h^k = b_h^k / H$.

        Compute Q function:

$$Q_h^k(\cdot, \cdot) \leftarrow \mathrm{Clip}_{[0, H-h+1]}\left(\mathrm{Clip}_{[0, H-h+1]}((w_h^k)^\top \phi_h(\cdot, \cdot) + r_h^k(\cdot, \cdot)) + b_h^k(\cdot, \cdot)\right),$$

        where $w_h^k = (\Lambda_h^k)^{-1} \sum_{i=1}^{k-1} \phi_h(s_h^i, a_h^i) \cdot V_{h+1}^k(s_{h+1}^i)$.

        Compute value function and policy:

$$V_h^k(\cdot) \leftarrow \max_{a \in \mathcal{A}} Q_h^k(\cdot, a), \pi_h^k(\cdot) \leftarrow \arg\max_{a \in \mathcal{A}} Q_h^k(\cdot, a).$$

    **end for**

    Collect a trajectory $\tau^k = (s_h^k, a_h^k, s_{h+1}^k)_{h=1}^H$ by running $\pi^k = \{\pi_h^k\}_{h=1}^H$ and add $\tau^k$ into $\mathcal{D}_{\mathrm{ex}}$.

**end for**

Sample $K$ states from the initial states $\{s_1^{i,\mathrm{in}}\}_{i=1}^K$ and add them to $\mathcal{D}_{\mathrm{in}}$.

Return $\mathcal{D}_{\mathrm{ex}}, \mathcal{D}_{\mathrm{in}}$.

---

- Third, our analysis of `REGIME-lin` differs from the one in the literature on linear MDPs. This is because while the style of our algorithm is akin to multi-policy evaluation in linear MDPs, the current literature of linear MDPs mainly focuses on Q-learning-style algorithms for policy optimization (Jin et al., 2019; Wang et al., 2020).

  More specifically, in `REGIME-lin`, we want to evaluate all the policies in a policy class under a certain reward function. This brings a new challenge that does not occur in Jin et al. (2019); Wang et al. (2020): our learning difficulty will scale with the size of the policy class . This forces us to find and analyze an appropriate policy class that is able to approximate the optimal policy while retaining a relatively small size. In addition, a key component of our analysis is the new concentration inequality in Lemma 13, which also differs from the existing concentration lemma in Jin et al. (2019); Wang et al. (2020) and may be of independent interest.

## B    OMIT DETAILS IN SECTION 4

In this section we present the details of Algorithm 4 and Algorithm 5. Here $\mathrm{Clip}_{[a,b]}(x)$ means $\min\{\max\{a, x\}, b\}$. In particular, when estimating $(\phi(\pi))_{h,j}$, we use the reward function $r_{h'}^{h,j}(s, a) = \phi_{h'}(s, a)^\top \theta_{h'}^{h,j}$ for all $h' \in [H]$ (up to an $R$ factor) where

$$\theta_{h'}^{h,j} = \begin{cases} \frac{1}{R} \cdot e_j, & \text{if} \quad h' = h, \\ 0, & \text{otherwise.} \end{cases}$$

Here $e_j$ is the one-hot vector whose $j$-th entry is 1. For simplicity, we denote $\widehat{V}^{r^{h,j},\pi}, \widehat{Q}^{r^{h,j},\pi}, w^{r^{h,j},\pi}$ by $V^{h,j,\pi}, Q^{h,j,\pi}, w^{h,j,\pi}$ and let the estimation $(\widehat{\phi}(\pi))_{h,j}$ be $R\widehat{V}^\pi(r^{h,j})$.

Then we have the following formal theorem characterizing the sample complexity of Algorithm 2:

**Theorem 4.** *Let*

$$\lambda_{\mathrm{ex}} = \lambda_{\mathrm{pl}} = R^2,$$

$$\beta_{\mathrm{ex}} = C_\beta dHR\sqrt{\log(dKHR/\delta)}, \beta_{\mathrm{pl}} = C_\beta dHR\sqrt{\log(dKHR\mathcal{N}_\Pi(\epsilon')/\delta)}$$

$$\lambda \geq 4HR^2, K \geq \widetilde{\mathcal{O}}\left(\frac{H^8 B^2 R^4 d^4 \log(\mathcal{N}_\Pi(\epsilon')/\delta)}{\epsilon^2}\right), N \geq \widetilde{\mathcal{O}}\left(\frac{\lambda\kappa^2 B^2 R^2 H^4 d^2 \log(1/\delta)}{\epsilon^2}\right),$$

---

**Algorithm 5** `REGIME-planning`

---

**Input**: Dataset $\mathcal{D}_{\text{ex}} = \{(s_h^i, a_h^i, s_{h+1}^i)\}_{i=1,h=1}^{K,H}, \mathcal{D}_{\text{in}} = \{s_1^{i,\text{in}}\}_{i=1}^K$, bonus parameter $\beta_{\text{pl}}$ and regularization parameter $\lambda_{\text{pl}}$, policy $\pi$, reward function $(r_h)_{h=1}^H$.

**for** $h' = H, \cdots, 1$ **do**

    Compute the covariance matrix: $\Lambda_{h'} \leftarrow \sum_{i=1}^K \phi_{h'}(s_{h'}^i, a_{h'}^i)\phi_{h'}(s_{h'}^i, a_{h'}^i)^\top + \lambda_{\text{pl}}I$.

    Compute the bonus: $b_{h'}(\cdot, \cdot) \leftarrow \min\{\beta_{\text{pl}}\|\phi_{h'}(\cdot, \cdot)\|_{(\Lambda_{h'})^{-1}}, 2(H - h + 1)\}$.

**end for**

Initialize: $\widehat{Q}_{H+1}^{r,\pi}(\cdot, \cdot) \leftarrow 0, \widehat{V}_{H+1}^{r,\pi}(\cdot) \leftarrow 0$.

**for** $h' = H, \cdots, 1$ **do**

    Compute Q function:

$$\widehat{Q}_{h'}^{r,\pi}(\cdot, \cdot) \leftarrow \text{Clip}_{[-(H-h+1), H-h+1]}\left(\text{Clip}_{[-(H-h+1), H-h+1]}((w_{h'}^{r,\pi})^\top \phi_{h'}(\cdot, \cdot) + r_{h'}(\cdot, \cdot)) + b_{h'}(\cdot, \cdot)\right),$$

    where $w_{h'}^{r,\pi} = (\Lambda_{h'})^{-1}\sum_{i=1}^K \phi_{h'}(s_{h'}^i, a_{h'}^i) \cdot \widehat{V}_{h'+1}^{r,\pi}(s_{h'+1}^i)$.

    Compute value function: $\widehat{V}_{h'}^{r,\pi}(\cdot) \leftarrow \mathbb{E}_{a\sim\pi_{h'}}\widehat{Q}_{h'}^{r,\pi}(\cdot, a)$.

**end for**

Compute $\widehat{V}^\pi(r) \leftarrow \frac{1}{K}\sum_{i=1}^K \widehat{V}_1^{r,\pi}(s_1^{i,\text{in}})$.

Return $\widehat{V}^\pi(r)$.

---

*where $\epsilon' = \frac{\epsilon}{72BR^2H\sqrt{d^H K^{H-1}}}$, $C_\beta > 0$ is a universal constant and $\kappa = 2 + \exp(2r_{\max}) + \exp(-2r_{\max})$. Then under Assumption 1 and 3, with probability at least $1 - \delta$, we have*

$$V^{r^*,\hat{\pi}} \geq V^{r^*,*} - \epsilon.$$

The proof is deferred to Appendix E.1.

## B.1 LOG-LINEAR POLICY CLASS

The sample complexity in Theorem 2 depends on the covering number of the policy class $\Pi$. Therefore we want to find a policy class for linear MDPs that is rich enough (i.e., contains near-global-optimal policies) while retains a small covering number at the same time. Indeed, the log-linear policy class (Agarwal et al., 2020b) satisfies this requirement, which is defined as follows:

$$\Pi = \left\{\pi : \pi_h^\zeta(a|s) = \frac{\exp(\zeta_h^\top \phi_h(s, a))}{\sum_{a'\in\mathcal{A}}\exp(\zeta_h^\top \phi_h(s, a'))}, \zeta_h \in \mathbb{B}(d, W), \forall s \in \mathcal{S}, a \in \mathcal{A}, h \in [H]\right\}$$

Here $\mathbb{B}(d, W)$ is the $d$-dimensional ball centered at the origin with radius $W$. The following proposition characterizes the covering number of such log-linear policy class:

**Proposition 1.** *Let $\Pi$ be the log-linear policy class. Then under Assumption 1, for any $\epsilon \leq 1$, we have $\log\mathcal{N}_\Pi(\epsilon) \leq Hd\log\left(\frac{12WR}{\epsilon}\right)$.*

Meanwhile, we can quantify the bias of such log-linear policy class as follows:

**Proposition 2.** *Let $W = \frac{(B+(H+\epsilon)\sqrt{d})H\log|\mathcal{A}|}{\epsilon}$, then under Assumption 1 and 3, we have*

$$V^{r^*,\pi_g} - \max_{\pi\in\Pi} V^{r^*,\pi} \leq \epsilon,$$

*where $\pi_g$ is the global optimal policy.*

Combining Theorem 4, Proposition 1 and Proposition 2, we know that the returned policy $\hat{\pi}$ by Algorithm 2 with log-linear policy classes can indeed compete against the global optimal policy with the following sample complexities:

$$N_{\text{tra}} = K + N = \widetilde{\mathcal{O}}\left(\frac{d^5 + \kappa^2 d^2}{\epsilon^2}\right), N_{\text{hum}} = \widetilde{\mathcal{O}}\left(\frac{\kappa^2 d^2}{\epsilon^2}\right).$$

## C  PROOF OF THEOREM 1 WITH KNOWN TRANSITIONS

In this section, we consider the proof of Theorem 1 when transitions are known, i.e., $\epsilon' = 0$ and $\widehat{P} = P^*$. In this case we have $\widehat{\phi}(\pi) = \phi(\pi)$. We will deal with the unknown transition in Appendix D.1.

First, note that from the definition of $\widehat{\pi}$, we have

$$V^{\widehat{r},\widehat{\pi}} \geq V^{\widehat{r},\pi^*},$$

where $\pi^*$ is the optimal policy with respect to the ground-truth reward $r^*$, i.e., $\pi^* = \arg\max_{\pi \in \Pi} V^{r^*,\pi}$. Therefore we can expand the suboptimality as follows:

$$
\begin{aligned}
V^{r^*,*} - V^{r^*,\widehat{\pi}} &= (V^{r^*,*} - V^{\widehat{r},\pi^*}) + (V^{\widehat{r},\pi^*} - V^{\widehat{r},\widehat{\pi}}) + (V^{\widehat{r},\widehat{\pi}} - V^{r^*,\widehat{\pi}}) \\
&\leq (V^{r^*,*} - V^{\widehat{r},\pi^*}) + (V^{\widehat{r},\widehat{\pi}} - V^{r^*,\widehat{\pi}}) \\
&= \mathbb{E}_{\tau \sim (\pi^*, P^*)}[\langle \phi(\tau), \theta^* - \widehat{\theta}\rangle] - \mathbb{E}_{\tau \sim (\widehat{\pi}, P^*)}[\langle \phi(\tau), \theta^* - \widehat{\theta}\rangle] \\
&= \langle \phi(\pi^*) - \phi(\widehat{\pi}), \theta^* - \widehat{\theta}\rangle \\
&\leq \|\phi(\pi^*) - \phi(\widehat{\pi})\|_{\Sigma_{N+1}^{-1}} \cdot \|\theta^* - \widehat{\theta}\|_{\Sigma_{N+1}},
\end{aligned}
\tag{3}
$$

where $\Sigma_n := \lambda I + \sum_{i=1}^{n-1}(\phi(\pi^{i,0}) - \phi(\pi^{i,1}))(\phi(\pi^{i,0}) - \phi(\pi^{i,1}))^\top$ for all $n \in [N+1]$. Here the third step is due to the definition of value function and the last step comes from Cauchy-Schwartz inequality. Next we will bound $\|\phi(\pi^*) - \phi(\widehat{\pi})\|_{\Sigma_{N+1}^{-1}}$ and $\|\theta^* - \widehat{\theta}\|_{\Sigma_{N+1}}$ respectively.

First for $\|\phi(\pi^*) - \phi(\widehat{\pi})\|_{\Sigma_{N+1}^{-1}}$, notice that $\Sigma_{N+1} \succeq \Sigma_n$ for all $n \in [N+1]$, which implies

$$
\begin{aligned}
\|\phi(\pi^*) - \phi(\widehat{\pi})\|_{\Sigma_{N+1}^{-1}} &\leq \frac{1}{N}\sum_{n=1}^{N}\|\phi(\pi^*) - \phi(\widehat{\pi})\|_{\Sigma_n^{-1}} \leq \frac{1}{N}\sum_{n=1}^{N}\|\phi(\pi^{n,0}) - \phi(\pi^{n,1})\|_{\Sigma_n^{-1}} \\
&\leq \frac{1}{\sqrt{N}}\sqrt{\sum_{n=1}^{N}\|\phi(\pi^{n,0}) - \phi(\pi^{n,1})\|_{\Sigma_n^{-1}}^2},
\end{aligned}
\tag{4}
$$

where the second step comes from the definition of $\pi^{n,0}$ and $\pi^{n,1}$ and the last step is due to Cauchy-Schwartz inequality. To bound the right hand side of (4), we utilize the following Elliptical Potential Lemma:

**Lemma 1** (Elliptical Potential Lemma). *For any $\lambda \geq R_x^2$ and $d \geq 1$, consider a sequence of vectors $\{x^n \in \mathbb{R}^d\}_{n=1}^N$ where $\|x^n\| \leq R_x$ for all $n \in [N]$. Let $\Sigma_n = \lambda I + \sum_{i=1}^{n-1} x^n (x^n)^\top$, then we have*

$$\sum_{n=1}^{N}\|x^n\|_{\Sigma_n^{-1}}^2 \leq 2d\log\left(1 + \frac{N}{d}\right).$$

The proof is deferred to Appendix C.1. Since we have $\lambda \geq 4HR^2$, by Lemma 1 we know

$$\sqrt{\sum_{n=1}^{N}\|\phi(\pi^{n,0}) - \phi(\pi^{n,1})\|_{\Sigma_n^{-1}}^2} \leq \sqrt{2HdN\log(1 + N/(Hd))}.$$

Combining the above inequality with (4), we have

$$\|\phi(\pi^*) - \phi(\widehat{\pi})\|_{\Sigma_{N+1}^{-1}} \leq \sqrt{\frac{2Hd\log(1 + N/(Hd))}{N}}.
\tag{5}$$

For $\|\theta^* - \widehat{\theta}\|_{\Sigma_{N+1}}$, first note that $\widehat{\theta}$ is the MLE estimator. Let $\widetilde{\Sigma}_n$ denote the empirical cumulative covariance matrix $\lambda I + \sum_{i=1}^{n-1}(\phi(\tau^{i,0}) - \phi(\tau^{i,1}))(\phi(\tau^{i,0}) - \phi(\tau^{i,1}))^\top$, then from the literature (Zhu et al., 2023), we know that MLE has the following guarantee:

**Lemma 2** (MLE guarantee). *For any $\lambda > 0$ and $\delta \in (0, 1)$, with probability at least $1 - \delta$, we have*

$$\|\hat{\theta} - \theta^*\|_{\widetilde{\Sigma}_{N+1}} \leq C_{\text{MLE}} \cdot \sqrt{\kappa^2(Hd + \log(1/\delta)) + \lambda H B^2}, \tag{6}$$

*where $\kappa = 2 + \exp(2r_{\max}) + \exp(-2r_{\max})$ and $C_{\text{MLE}} > 0$ is a universal constant.*

The proof is deferred to Appendix C.2. With Lemma 2, to $\|\theta^* - \hat{\theta}\|_{\Sigma_{N+1}}$ we only need to show $\widetilde{\Sigma}_{N+1}$ is close to $\Sigma_{N+1}$. This can be achieved by the following concentration result from the literature:

**Lemma 3** ([Pacchiano et al. (2021)][Lemma 7]). *For any $\lambda > 0$ and $\delta \in (0, 1)$, with probability at least $1 - \delta$, we have*

$$\|\theta^* - \hat{\theta}\|_{\Sigma_{N+1}}^2 \leq 2\|\theta^* - \hat{\theta}\|_{\widetilde{\Sigma}_{N+1}}^2 + C_{\text{CON}} H^3 dR^2 B^2 \log(N/\delta), \tag{7}$$

*where $C_{\text{CON}} > 0$ is a universal constant.*

Therefore, combining (7) and (6), by union bound with probability at least $1 - \delta$, we have that

$$\|\theta^* - \hat{\theta}\|_{\Sigma_{N+1}} \leq C_1 \cdot \kappa B R \sqrt{\lambda H^3 d \log(N/\delta)}, \tag{8}$$

where $C_1$ is a universal constant.

Thus substituting (5) and (8) into (3), we have $V^*(r^*) - V(r^*, \hat{\pi}) \leq \epsilon$ with probability at least $1 - \delta$ as long as

$$N \geq \widetilde{\mathcal{O}}\Big(\frac{\lambda \kappa^2 B^2 R^2 H^4 d^2 \log(1/\delta)}{\epsilon^2}\Big).$$

## C.1 PROOF OF LEMMA 1

Note that when $\lambda \geq R_x^2$, we have $\|x^n\|_{\Sigma_n^{-1}} \leq 1$ for all $n \in [N]$, which implies that for all $n \in [N]$, we have

$$\|x^n\|_{\Sigma_n^{-1}}^2 \leq \log\left(1 + \|x^n\|_{\Sigma_n^{-1}}^2\right).$$

On the other hand, let $w^n$ denote $\|x^n\|_{\Sigma_n^{-1}}$, then we know for any $n \in [N-1]$

$$\begin{aligned}
\log \det \Sigma_{n+1} = \det(\Sigma_n + x^n(x^n)^\top) &= \log \det(\Sigma_n^{1/2}(I + \Sigma_n^{-1/2}x^n(x^n)^\top\Sigma_n^{-1/2})\Sigma_n^{1/2}) \\
&= \log \det(\Sigma_n) + \log \det(I + (\Sigma_n^{-1/2}x^n)(\Sigma_n^{-1/2}x^n)^\top) \\
&= \log \det(\Sigma_n) + \log \det(I + (\Sigma_n^{-1/2}x^n)^\top(\Sigma_n^{-1/2}x^n)) \\
&= \log \det(\Sigma_n) + \log\left(1 + \|x^n\|_{\Sigma_n^{-1}}^2\right),
\end{aligned}$$

where the fourth step is due to the property of determinants. Therefore we have

$$\sum_{n=1}^N \log\left(1 + \|x^n\|_{\Sigma_n^{-1}}^2\right) = \log \det \Sigma_{N+1} - \log \det \Sigma_1 = \log(\det \Sigma_{N+1}/\det \Sigma_1)$$

$$= \log \det\left(I + \frac{1}{\lambda}\sum_{n=1}^N x^n(x^n)^\top\right).$$

Now let $\{\lambda_i\}_{i=1}^d$ denote the eigenvalues of $\sum_{n=1}^N x^n(x^n)^\top$, then we know

$$\begin{aligned}
\log \det\left(I + \frac{1}{\lambda}\sum_{n=1}^N x^n(x^n)^\top\right) &= \log\left(\prod_{i=1}^d (1 + \lambda_i/\lambda)\right) \\
&\leq d\log\left(\frac{1}{d}\sum_{i=1}^d (1 + \lambda_i/\lambda)\right) \leq d\log\left(1 + \frac{NR_x^2}{d\lambda}\right) \leq d\log\left(1 + \frac{N}{d}\right),
\end{aligned}$$

where the third step comes from $\sum_{i=1}^d \lambda_i = \text{Tr}\left(\sum_{n=1}^N x^n(x^n)^\top\right) = \sum_{n=1}^N \|x^n\|^2 \leq NR_x^2$ and the last step is due to the fact that $\lambda \geq R_x^2$. This concludes our proof.

## C.2    PROOF OF LEMMA 2

First note that we have the following lemma from literature:

**Lemma 4** ( (Zhu et al., 2023)[Lemma 3.1]). *For any $\lambda' > 0$, with probability at least $1 - \delta$, we have*

$$\|\widehat{\theta} - \theta^*\|_{D+\lambda' I} \leq O\left(\sqrt{\frac{\kappa^2(Hd + \log(1/\delta))}{N}} + \lambda' HB^2\right),$$

*where $D = \frac{1}{N}\sum_{i=1}^{N}(\phi(\tau^{i,0}) - \phi(\tau^{i,1}))(\phi(\tau^{i,0}) - \phi(\tau^{i,1}))^\top$.*

Therefore let $\lambda' = \frac{\lambda}{N}$ and from the above lemma we can obtain

$$\|\widehat{\theta} - \theta^*\|_{\frac{\widetilde{\Sigma}_{N+1}}{N}} \leq O\left(\sqrt{\frac{\kappa^2(Hd + \log(1/\delta))}{N}} + \frac{\lambda HB^2}{N}\right),$$

which is equivalent to

$$\|\widehat{\theta} - \theta^*\|_{\widetilde{\Sigma}_{N+1}} \leq O\left(\sqrt{\kappa^2(Hd + \log(1/\delta)) + \lambda HB^2}\right).$$

This concludes our proof.

# D    PROOFS IN SECTION 3

## D.1    PROOF OF THEOREM 1

Note that from the proof of Theorem 1 with known transition dynamics, we have:

$$V^{r^*,*} - V^{r^*,\widehat{\pi}} \leq \langle \phi(\pi^*) - \phi(\widehat{\pi}), \theta^* - \widehat{\theta} \rangle + (V^{\widehat{r},\pi^*} - V^{\widehat{r},\widehat{\pi}}), \tag{9}$$

Then we have

$$V^{r^*,*} - V^{r^*,\widehat{\pi}} \leq \langle \phi(\pi^*) - \widehat{\phi}(\pi^*), \theta^* - \hat{\theta} \rangle + \langle \widehat{\phi}(\hat{\pi}) - \phi(\hat{\pi}), \theta^* - \hat{\theta} \rangle$$
$$+ \langle \widehat{\phi}(\pi^*) - \widehat{\phi}(\hat{\pi}), \theta^* - \hat{\theta} \rangle + (V^{\widehat{r},\pi^*} - V^{\widehat{r},\widehat{\pi}}). \tag{10}$$

Now we only need to bound the three terms in the RHS of (10). For the first and second term, we need to utilize the following lemma:

**Lemma 5.** *Let $d_h^\pi(s, a)$ and $\widehat{d}_h^\pi(s, a)$ denote the visitation measure of policy $\pi$ under $P^*$ and $\hat{P}$. Then with probability at least $1 - \delta/4$, we have for all $h \in [H]$ and $\pi \in \Pi$,*

$$\|d_h^\pi - \widehat{d}_h^\pi\|_1 \leq h\epsilon'. \tag{11}$$

Let $\mathcal{E}_1$ denote the event when (11) holds. Then under event $\mathcal{E}_1$, we further have the following lemma:

**Lemma 6.** *Under event $\mathcal{E}_1$, for all policy $\pi \in \Pi$ and vector $v = [v_1, \cdots, v_H]$ where $v_h \in \mathbb{R}^d$ and $\|v_h\| \leq 2B$ for all $h \in [H]$ we have,*

$$|\langle \phi(\pi) - \widehat{\phi}(\pi), v \rangle| \leq BRH^2\epsilon'.$$

Substitute Lemma 6 into (10), we have

$$V^{r^*,*} - V^{r^*,\widehat{\pi}} \leq \langle \widehat{\phi}(\pi^*) - \widehat{\phi}(\widehat{\pi}), \theta^* - \widehat{\theta} \rangle + 2BRH^2\epsilon' + (V^{\widehat{r},\pi^*} - V^{\widehat{r},\widehat{\pi}}).$$

Then by Cauchy-Schwartz inequality, we have under event $\mathcal{E}_1$,

$$V^{r^*,*} - V^{r^*,\widehat{\pi}} \leq \|\widehat{\phi}(\pi^*) - \widehat{\phi}(\widehat{\pi})\|_{\widehat{\Sigma}_{N+1}^{-1}} \cdot \|\theta^* - \widehat{\theta}\|_{\widehat{\Sigma}_{N+1}} + 2BRH^2\epsilon' + (V^{\widehat{r},\pi^*} - V^{\widehat{r},\widehat{\pi}}). \tag{12}$$

Following the same analysis in the proof of Theorem 1 with known transition, we know

$$\|\widehat{\phi}(\pi^*) - \widehat{\phi}(\widehat{\pi})\|_{\widehat{\Sigma}_{N+1}^{-1}} \leq \sqrt{\frac{2Hd\log(1 + N/(Hd))}{N}}. \tag{13}$$

Now we only need to bound $\|\theta^* - \widehat{\theta}\|_{\widehat{\Sigma}_{N+1}}$. Similar to the proof of Theorem 1 with known transition, we use $\Sigma_n$ and $\widetilde{\Sigma}_n$ to denote $\lambda I + \sum_{i=1}^{n-1}(\phi(\pi^{i,0}) - \phi(\pi^{i,1}))(\phi(\pi^{i,0}) - \phi(\pi^{i,1}))^\top$ and $\lambda I + \sum_{i=1}^{n-1}(\phi(\tau^{i,0}) - \phi(\tau^{i,1}))(\phi(\tau^{i,0}) - \phi(\tau^{i,1}))^\top$ respectively. Then under event $\mathcal{E}_1$, we have the following connection between $\widehat{\Sigma}_{N+1}$ and $\Sigma_{N+1}$:

**Lemma 7.** *Under event $\mathcal{E}_1$, we have*

$$\|\theta^* - \widehat{\theta}\|_{\widehat{\Sigma}_{N+1}} \leq \sqrt{2}\|\theta^* - \widehat{\theta}\|_{\Sigma_{N+1}} + 2\sqrt{2N}BRH^2\epsilon'.$$

Combining Lemma 7 with Lemma 2 and Lemma 3, we have under event $\mathcal{E}_1 \cap \mathcal{E}_2$,

$$\|\theta^* - \widehat{\theta}\|_{\widehat{\Sigma}_{N+1}} \leq \sqrt{2}\|\theta^* - \widehat{\theta}\|_{\Sigma_{N+1}} + 2\sqrt{2N}BRH^2\epsilon'$$
$$\leq C_2 \cdot \kappa BR\sqrt{\lambda H^3 d \log(N/\delta)} + 2\sqrt{2N}BRH^2\epsilon', \tag{14}$$

where $\Pr(\mathcal{E}_2) \geq 1 - \delta/2$ and $C_2 > 0$ is a universal constant.

Now we only need to bound $(V^{\widehat{r},\pi^*} - V^{\widehat{r},\widehat{\pi}})$, which can be achieved with Lemma 6:

$$V^{\widehat{r},\pi^*} - V^{\widehat{r},\widehat{\pi}} = \langle \phi(\pi^*), \widehat{\theta} \rangle - \langle \phi(\widehat{\pi}), \widehat{\theta} \rangle$$
$$= \langle \phi(\pi^*) - \widehat{\phi}(\pi^*), \widehat{\theta} \rangle + \langle \widehat{\phi}(\pi^*) - \widehat{\phi}(\widehat{\pi}), \widehat{\theta} \rangle + \langle \widehat{\phi}(\widehat{\pi}) - \phi(\widehat{\pi}), \widehat{\theta} \rangle \leq 2BRH^2\epsilon', \tag{15}$$

where the last step comes from Lemma 6 and the definition of $\widehat{\pi}$.

Combining (12), (13) (14) and (15), we have $V^{r^*,*} - V^{r^*,\widehat{\pi}} \leq \epsilon$ with probability at least $1 - \delta$ as long as

$$\epsilon' \leq \frac{\epsilon}{6BRH^2}, \qquad N \geq \widetilde{\mathcal{O}}\Big(\frac{\lambda\kappa^2 B^2 R^2 H^4 d^2 \log(1/\delta)}{\epsilon^2}\Big).$$

### D.2 PROOF OF LEMMA 5

First notice that $d_h^\pi(s,a) = d_h^\pi(s)\pi(a|s)$ and $\widehat{d}_h^\pi(s,a) = \widehat{d}_h^\pi(s)\pi(a|s)$, which implies that for all $h \in [H]$

$$\big\|d_h^\pi - \widehat{d}_h^\pi\big\|_1 = \sum_{s,a}\big|d_h^\pi(s,a) - \widehat{d}_h^\pi(s,a)\big| = \sum_{s,a}\big|d_h^\pi(s) - \widehat{d}_h^\pi(s)\big|\pi(a|s)$$
$$= \sum_s \big|d_h^\pi(s) - \widehat{d}_h^\pi(s)\big|\sum_a \pi(a|s) = \sum_s \big|d_h^\pi(s) - \widehat{d}_h^\pi(s)\big|.$$

Therefore we only need to prove $\sum_s \big|d_h^\pi(s) - \widehat{d}_h^\pi(s)\big| \leq h\epsilon'$ for all $h \in [H]$. We use induction to prove this. First for the base case, we have $\sum_s |d_1^\pi(s) - \widehat{d}_1^\pi(s)| = \sum_s \big|P_1^*(s) - \widehat{P}_1(s)\big| \leq \epsilon'$ according to the guarantee of the reward-free learnign oracle $\mathcal{P}$.

Now assume that $\sum_s \big|d_{h'}^\pi(s) - \widehat{d}_{h'}^\pi(s)\big| \leq h'\epsilon'$ for all $h' \in [h]$ where $h \in [H-1]$. Then we have

$$\sum_s \big|d_{h+1}^\pi(s) - \widehat{d}_{h+1}^\pi(s)\big| = \sum_s \Big|\sum_{s',a'}\widehat{d}_h^\pi(s')\pi(a'|s')\widehat{P}_h(s|s',a') - d_h^\pi(s')\pi(a'|s')P_h^*(s|s',a')\Big|$$
$$\leq \Big(\sum_{s,s',a'}\big|\widehat{d}_h^\pi(s') - d_h^\pi(s')\big|\pi(a'|s')\widehat{P}_h(s|s',a')\Big)$$
$$+ \Big(\sum_{s,s',a'}d_h^\pi(s')\pi(a'|s')\big|\widehat{P}_h(s|s',a') - P_h^*(s|s',a')\big|\Big)$$
$$= \Big(\sum_{s'}\big|\widehat{d}_h^\pi(s') - d_h^\pi(s')\big|\sum_{a'}\pi(a'|s')\sum_s \widehat{P}_h(s|s',a')\Big)$$
$$+ \mathbb{E}_{\pi,P^*}[\|\widehat{P}_h(\cdot|s',a') - P_h^*(\cdot|s',a')\|_1]$$
$$\leq (h+1)\epsilon',$$

where the second step comes from the triangle inequality and the last step is dueto the induction hypothesis and the guarantee of $\mathcal{P}$. Therefore, we have $\sum_s |d_{h+1}^\pi(s) - \widehat{d}_{h+1}^\pi(s)| \leq (h+1)\epsilon'$. Then by induction, we know $\sum_s |d_h^\pi(s) - \widehat{d}_h^\pi(s)| \leq h\epsilon'$ for all $h \in [H]$, which concludes our proof.

### D.3 PROOF OF LEMMA 6

Note that from the definition of $\phi(\pi)$ we have

$$\langle \phi(\pi), v \rangle = \mathbb{E}_{\tau \sim (\pi, P^*)} \left[ \sum_{h=1}^{H} \phi_h^\top(s_h, a_h) v_h \right] = \sum_{h=1}^{H} \sum_{s_h, a_h} d_h^\pi(s_h, a_h) \phi_h^\top(s_h, a_h) v_h.$$

Similarly, we have

$$\langle \widehat{\phi}(\pi), v \rangle = \sum_{h=1}^{H} \sum_{s_h, a_h} \widehat{d}_h^\pi(s_h, a_h) \phi_h^\top(s_h, a_h) v_h.$$

Therefore,

$$|\langle \phi(\pi) - \widehat{\phi}(\pi), v \rangle| \le \sum_{h=1}^{H} \sum_{s_h, a_h} |\widehat{d}_h^\pi(s_h, a_h) - d_h^\pi(s_h, a_h)| \cdot |\phi_h^\top(s_h, a_h) v_h|$$

$$\le 2BR \sum_{h=1}^{H} \sum_{s_h, a_h} |\widehat{d}_h^\pi(s_h, a_h) - d_h^\pi(s_h, a_h)|$$

$$\le 2BR \sum_{h=1}^{H} h\epsilon' \le BRH^2\epsilon',$$

where the first step is due to the triangle inequality and the third step comes from Lemma 5. This concludes our proof.

### D.4 PROOF OF LEMMA 7

We use $\Delta\theta$ to denote $\theta^* - \hat{\theta}$ in this proof. From Lemma 6, we know that for any policy $\pi$,

$$|\langle \phi(\pi) - \widehat{\phi}(\pi), \Delta\theta \rangle| \le BRH^2\epsilon'.$$

By the triangle inequality, this implies that for any policy $\pi^0, \pi^1$,

$$|\langle \widehat{\phi}(\pi^0) - \widehat{\phi}(\pi^1), \Delta\theta \rangle| \le |\langle \phi(\pi^0) - \phi(\pi^1), \Delta\theta \rangle| + 2BRH^2\epsilon'.$$

Therefore we have for any policy $\pi^0, \pi^1$,

$$|\langle \widehat{\phi}(\pi^0) - \widehat{\phi}(\pi^1), \Delta\theta \rangle|^2 \le 2|\langle \phi(\pi^0) - \phi(\pi^1), \Delta\theta \rangle|^2 + 8(BRH^2\epsilon')^2. \tag{16}$$

Note that from the definition of $\widehat{\Sigma}_{N+1}$ and $\Sigma_{N+1}$, we have

$$\|\Delta\theta\|_{\widehat{\Sigma}_{N+1}}^2 = \Delta\theta^\top \left( \lambda I + \sum_{n=1}^{N} (\widehat{\phi}(\pi^{n,0}) - \widehat{\phi}(\pi^{n,1}))(\widehat{\phi}(\pi^{n,0}) - \widehat{\phi}(\pi^{n,1}))^\top \right) \Delta\theta$$

$$= \lambda\|\Delta\theta\|^2 + \sum_{n=1}^{N} |\langle \widehat{\phi}(\pi^{n,0}) - \widehat{\phi}(\pi^{n,1}), \Delta\theta \rangle|^2$$

$$\le 2 \left( \lambda\|\Delta\theta\|^2 + \sum_{n=1}^{N} |\langle \phi(\pi^{n,0}) - \phi(\pi^{n,1}), \Delta\theta \rangle|^2 \right) + 8N(BRH^2\epsilon')^2$$

$$= 2\|\Delta\theta\|_{\Sigma_{N+1}}^2 + 8N(BRH^2\epsilon')^2,$$

where the third step comes from (16). This implies that

$$\|\Delta\theta\|_{\widehat{\Sigma}_{N+1}} \le \sqrt{2}\|\Delta\theta\|_{\Sigma_{N+1}} + 2\sqrt{2N}BRH^2\epsilon',$$

which concludes our proof.

# E   PROOFS IN SECTION 4 AND APPENDIX B

## E.1   PROOF OF THEOREM 4

First note that Algorithm 4 provides us with the following guarantee:

**Lemma 8.** *We have with probability at least $1 - \delta/6$ that*

$$\mathbb{E}_{s_1 \sim P_1(\cdot)}[V_1^{b/H,*}(s_1)] \leq C_{\text{lin}}\sqrt{d^3H^4R^2 \cdot \log(\mathcal{N}_{\Pi}(\epsilon')dKHR/\delta)/K},$$

*where $b_h$ is defined in Algorithm 5 and $C_{\text{lin}} > 0$ is a universal constant. Here $V_1^{r,*}(s_1) := \max_{\pi \in \Pi} V_1^{r,\pi}(s_1)$.*

Lemma 8 is adapted from Wang et al. (2020)[Lemma 3.2] and we highlight the difference of the proof in Appendix E.2. Then we consider a $\epsilon'$-covering for $\Pi$, denoted by $\mathcal{C}(\Pi, \epsilon')$. Following the similar analysis in Wang et al. (2020)[Lemma 3.3], we have the following lemma:

**Lemma 9.** *With probability $1 - \delta/6$, for all $h' \in [H]$, policy $\pi \in \mathcal{C}(\Pi, \epsilon')$ and linear reward function $r$ with $r_h \in [-1, 1]$, we have*

$$Q_{h'}^{r,\pi}(\cdot,\cdot) \leq \widehat{Q}_{h'}^{r,\pi}(\cdot,\cdot) \leq r_{h'}(\cdot,\cdot) + \sum_{s'} P_{h'}^*(s'|\cdot,\cdot)\widehat{V}_{h'+1}^{r,\pi}(s') + 2b_{h'}(\cdot,\cdot).$$

The proof of Lemma 9 is deferred to Appendix E.3. Denote the event in Lemma 8 and Lemma 9 by $\mathcal{E}_4$ and $\mathcal{E}_5$ respectively. Then under event $\mathcal{E}_4 \cap \mathcal{E}_5$, we have for all policy $\pi \in \mathcal{C}(\Pi, \epsilon')$ and all linear reward function $r$ with $r_h \in [-1, 1]$,

$$0 \leq \mathbb{E}_{s_1 \sim P_1^*(\cdot)}[\widehat{V}_1^{r,\pi}(s_1) - V_1^{r,\pi}(s_1)] \leq 2\mathbb{E}_{s_1 \sim P_1^*(\cdot)}[V_1^{b,\pi}(s_1)]$$

$$\leq 2H\mathbb{E}_{s_1 \sim P_1(\cdot)}[V_1^{b/H,*}(s_1)] \leq 2C_{\text{lin}}\sqrt{\frac{d^3H^6R^2 \cdot \log(dKHR\mathcal{N}_{\Pi}(\epsilon')/\delta)}{K}} \leq \epsilon_0, \quad (17)$$

where $\epsilon_0 = \frac{\epsilon}{72BR\sqrt{Hd}}$. Here the first step comes from the left part of Lemma 9 and the second step is due to the right part of Lemma 9.

Note that in the proof of Lemma 13, we calculate the covering number of the function class $\{\widehat{V}_1^{r,\pi} : r$ is linear and $r_h \in [-1, 1]\}$ for any fixed $\pi$ in (24). Then by Azuma-Hoeffding's inequality and (24), we have with probability at least $1 - \delta/6$ that for all policy $\pi \in \mathcal{C}(\Pi, \epsilon')$ and all linear reward function $r$ with $r_h \in [-1, 1]$ that

$$\left|\mathbb{E}_{s_1 \sim P_1^*(\cdot)}[\widehat{V}_1^{r,\pi}(s_1)] - \frac{1}{K}\sum_{i=1}^{K}\widehat{V}_1^{r,\pi}(s_1^{i,\text{in}})\right| \leq C_3H \cdot \sqrt{\frac{\log(\mathcal{N}_{\Pi}(\epsilon')HKdR/\delta)}{K}} \leq \epsilon_0, \quad (18)$$

where $C_3 > 0$ is a universal constant.

Combining (17) and (18), we have with probability at least $1 - \delta/2$ that for all policy $\pi \in \mathcal{C}(\Pi, \epsilon')$ and all linear reward function $r$ with $r_h \in [-1, 1]$

$$|\widehat{V}^{\pi}(r) - V^{r,\pi}| \leq 2\epsilon_0. \quad (19)$$

This implies that we can estimate the value function for all $\pi \in \mathcal{C}(\Pi, \epsilon')$ and linear reward function $r$ with $r_h \in [-1, 1]$ up to estimation error $2\epsilon_0$.

Now we consider any policy $\pi \in \Pi$. Suppose that $\pi' \in \mathcal{C}(\Pi, \epsilon')$ satisfies that

$$\max_{s \in \mathcal{S}, h \in [H]} \|\pi_h(\cdot|s) - \pi'_h(\cdot|s)\|_1 \leq \epsilon'. \quad (20)$$

Then we can bound $|\widehat{V}^{\pi}(r) - \widehat{V}^{\pi'}(r)|$ and $|V^{r,\pi} - V^{r,\pi'}|$ for all linear reward function $r$ with $r_h \in [-1, 1]$ respectively.

For $|V^{r,\pi} - V^{r,\pi'}|$, note that we have the following performance difference lemma:

**Lemma 10.** *For any policy $\pi, \pi'$ and reward function $r$, we have*

$$V^{r,\pi'} - V^{r,\pi} = \sum_{h=1}^{H} \mathbb{E}_{\pi',P^*}\Big[\langle Q_h^{r,\pi}(s_h,\cdot), \pi'_h(\cdot|s) - \pi_h(\cdot|s)\rangle\Big].$$

The proof is deferred to Appendix E.4. Therefore from Lemma 10 we have

$$
|V^{r,\pi'} - V^{r,\pi}| \le \sum_{h'=1}^{H} \mathbb{E}_{\pi,P^*} \Big[ \big| \langle Q_{h'}^{r^{h,j},\pi'}(s_{h'},\cdot), \pi_{h'}(\cdot|s) - \pi'_{h'}(\cdot|s) \rangle \big| \Big]
$$

$$
\le \sum_{h'=1}^{H} \mathbb{E}_{\pi,P^*} \Big[ \| \pi_{h'}(\cdot|s) - \pi'_{h'}(\cdot|s) \|_1 \Big] \le H\epsilon'. \tag{21}
$$

On the other hand, we have the following lemma to bound $|\widehat{V}^\pi(r) - \widehat{V}^{\pi'}(r)|$:

**Lemma 11.** *Suppose (20) holds and $\widehat{V}^\pi(r), \widehat{V}^{\pi'}(r)$ are calculated as in Algorithm 5. Then for all linear reward function $r$ with $0 \le r(\tau) \le r_{\max}$, we have*

$$
|\widehat{V}^\pi(r) - \widehat{V}^{\pi'}(r)| \le \epsilon_{\mathrm{cover}} := \frac{H\epsilon'}{\sqrt{dK}-1} \cdot (dK)^{\frac{H}{2}}. \tag{22}
$$

The proof is deferred to Appendix E.5.

Combining (19),(21) and (22), we have for all policy $\pi \in \Pi$ and linear reward function $r$ with $0 \le r(\tau) \le r_{\max}$,

$$
|\widehat{V}^\pi(r) - V^{r,\pi}| \le 2\epsilon_0 + H\epsilon' + \epsilon_{\mathrm{cover}}. \tag{23}
$$

In particular, since $(\phi(\pi))_{h,j} = RV^{r^{h,j},\pi}$, we have for all policy $\pi \in \Pi$ and $h \in [H], j \in [d]$,

$$
|(\phi(\pi))_{h,j} - (\widehat{\phi}(\pi))_{h,j}| \le 2R\epsilon_0 + HR\epsilon' + R\epsilon_{\mathrm{cover}}.
$$

This implies that for all policy $\pi \in \Pi$ and any $v$ defined in Lemma 6, we have

$$
|\langle (\phi(\pi)) - (\widehat{\phi}(\pi)), v \rangle| \le 2BH\sqrt{d}(2R\epsilon_0 + HR\epsilon' + R\epsilon_{\mathrm{cover}}).
$$

The rest of the proof is the same as Theorem 1 and thus is omitted here. The only difference is that we need to show $\widehat{\pi}$ is a near-optimal policy with respect to $\widehat{r}$. This can be proved as follows:

$$
V^{\widehat{r},*} - V^{\widehat{r},\widehat{\pi}} = \Big( V^{\widehat{r},*} - \widehat{V}^{\widehat{\pi}}(\widehat{r}) \Big) + \Big( \widehat{V}^{\widehat{\pi}}(\widehat{r}) - \widehat{V}^{\pi^*(\widehat{r})}(\widehat{r}) \Big) + \Big( \widehat{V}^{\pi^*(\widehat{r})}(\widehat{r}) - V^{\widehat{r},\widehat{\pi}} \Big)
$$

$$
\le 4\epsilon_0 + 2H\epsilon' + 2\epsilon_{\mathrm{cover}},
$$

where the last step comes from (23) and the definition of $\widehat{\pi}$.

## E.2    Proof of Lemma 8

Here we outline the difference of the proof from Wang et al. (2020)[Lemma 3.2]. First, we also have the following concentration guarantee:

**Lemma 12.** *Fix a policy $\pi$. Then with probability at least $1 - \delta$, we have for all $h \in [H]$ and $k \in [K]$,*

$$
\left\| \sum_{i=1}^{k} \phi_h^i \Big( V_{h+1}^k(s_{h+1}^i) - \sum_{s' \in \mathcal{S}} P_h^*(s'|s_h^i, a_h^i) V_{h+1}^k(s') \Big) \right\|_{\Lambda_h^{-1}} \le O\big( dHR\sqrt{\log(dKHR/\delta)} \big).
$$

The proof is almost the same as Lemma 13 and thus is omitted here. Then following the same arguments in Wang et al. (2020), we have the following inequality under Lemma 12:

$$
\left| \phi_h(s,a)^\top w_h^k - \sum_{s' \in \mathcal{S}} P_h^*(s'|s,a) V_{h+1}^k(s') \right| \le \beta_{\mathrm{ex}} \| \phi_h(s,a) \|_{(\Lambda_h^k)^{-1}}.
$$

Note that $V_{h+1}^k(s) \in [0, H-h]$ for all $s \in \mathcal{S}$, which implies that

$$
0 \le \sum_{s' \in \mathcal{S}} P_h^*(s'|s,a) V_{h+1}^k(s') + r_h^k(s,a) \le H - h + 1.
$$

Note that Clip is a contraction operator, which implies that

$$\left| \text{Clip}_{[0,H-h+1]}((w_h^k)^\top \phi_h(s,a) + r_h^k(s,a)) - \left( \sum_{s' \in \mathcal{S}} P_h^*(s'|s,a)V_{h+1}^k(s') + r_h^k(s,a) \right) \right|$$

$$\leq \left| (w_h^k)^\top \phi_h(s,a) - \sum_{s' \in \mathcal{S}} P_h^*(s'|s,a)V_{h+1}^k(s') \right| \leq \beta_{\text{ex}} \|\phi_h(s,a)\|_{(\Lambda_h^k)^{-1}}.$$

On the other hand,

$$\left| \text{Clip}_{[0,H-h+1]}((w_h^k)^\top \phi_h(s,a) + r_h^k(s,a)) - \left( \sum_{s' \in \mathcal{S}} P_h^*(s'|s,a)V_{h+1}^k(s') + r_h^k(s,a) \right) \right| \leq H - h + 1.$$

This implies that

$$\left| \text{Clip}_{[0,H-h+1]}((w_h^k)^\top \phi_h(s,a) + r_h^k(s,a)) - \left( \sum_{s' \in \mathcal{S}} P_h^*(s'|s,a)V_{h+1}^k(s') + r_h^k(s,a) \right) \right| \leq b_h^k(s,a).$$

The rest of the proof is the same as Wang et al. (2020) and thus is omitted.

### E.3 PROOF OF LEMMA 9

In the following discussion we will use $\phi_h^i$ to denote $\phi_h(s_h^i, a_h^i)$. First we need the following concentration lemma which is similar to Jin et al. (2020b)[Lemma B.3]:

**Lemma 13.** *Fix a policy $\pi$. Then with probability at least $1 - \delta$, we have for all $h \in [H]$ and linear reward functions $r$ with $r_h \in [-1, 1]$,*

$$\left\| \sum_{i=1}^K \phi_h^i \left( \widehat{V}_{h+1}^{r,\pi}(s_{h+1}^i) - \sum_{s' \in \mathcal{S}} P_h^*(s'|s_h^i, a_h^i)\widehat{V}_{h+1}^{r,\pi}(s') \right) \right\|_{\Lambda_h^{-1}} \leq O\big(dHR\sqrt{\log(dKHR/\delta)}\big).$$

The proof is deferred to Appendix E.6. Then by union bound, we know with probability $1 - \delta/6$, we have for all policy $\pi \in \mathcal{C}(\Pi, \epsilon')$, $h \in [H]$ and linear reward functions $r$ with $r_h \in [-1, 1]$ that

$$\left\| \sum_{i=1}^K \phi_h^i \left( \widehat{V}_{h+1}^{r,\pi}(s_{h+1}^i) - \sum_{s' \in \mathcal{S}} P_h^*(s'|s_h^i, a_h^i)\widehat{V}_{h+1}^{r,\pi}(s') \right) \right\|_{\Lambda_h^{-1}} \leq O\big(dHR\sqrt{\log(dKHR\mathcal{N}_\Pi(\epsilon')/\delta)}\big).$$

Let $\mathcal{E}_6$ denote the event thar the above inequality holds. Then under $\mathcal{E}_6$, following the same analysis in (Wang et al., 2020)[Lemma 3.1], we have for all policy $\pi \in \mathcal{C}(\Pi, \epsilon')$, $(s,a) \in \mathcal{S} \times \mathcal{A}$, $h \in [H]$ and linear reward functions $r$ with $r_h \in [-1, 1]$ that

$$\left| \phi_h(s,a)^\top w_h^{r,\pi} - \sum_{s' \in \mathcal{S}} P_h^*(s'|s,a)\widehat{V}_{h+1}^{r,\pi}(s') \right| \leq \beta_{\text{pl}} \|\phi_h(s,a)\|_{\Lambda_h^{-1}}.$$

Form the contraction property of Clip and the fact that $\sum_{s' \in \mathcal{S}} P_h^*(s'|s,a)\widehat{V}_{h+1}^{r,\pi}(s') + r_h(s,a) \in [-(H-h+1), H-h+1]$, we know

$$\left| \text{Clip}_{[-(H-h+1),H-h+1]}((w_h^{r,\pi})^\top \phi_h(s,a) + r_h(s,a)) - \sum_{s' \in \mathcal{S}} P_h^*(s'|s,a)\widehat{V}_{h+1}^{r,\pi}(s') - r_h(s,a) \right| \leq b_h(s,a)$$

Therefore, under $\mathcal{E}_6$ we have

$$\widehat{Q}_h^{r,\pi}(s,a) \leq r_h(s,a) + \sum_{s'} P_h^*(s'|s,a)\widehat{V}_{h+1}^{r,\pi}(s') + 2b_h(s,a).$$

Now we only need to prove under $\mathcal{E}_6$, for all policy $\pi \in \mathcal{C}(\Pi, \epsilon')$, $(s,a) \in \mathcal{S} \times \mathcal{A}$, $h \in [H]$ and linear reward function $r$ with $r_h \in [-1, 1]$, we have $Q_h^{r,\pi}(s,a) \leq \widehat{Q}_h^{r,\pi}(s,a)$. We use induction to prove this. The claim holds obviously for $h = H + 1$. Then we suppose for some $h \in [H]$, we have

$Q_{h+1}^{r,\pi}(s,a) \le \widehat{Q}_{h+1}^{r,\pi}(s,a)$ for all policy $\pi \in \mathcal{C}(\Pi, \epsilon')$, $(s,a) \in \mathcal{S} \times \mathcal{A}$ and linear reward function $r$ with $r_h \in [-1, 1]$. Then we have:

$$V_{h+1}^{r,\pi}(s) = \mathbb{E}_{a\sim\pi_{h+1}(\cdot|s)}\left[Q_{h+1}^{r,\pi}(s,a)\right] \le \widehat{V}_{h+1}^{r,\pi}(s) = \mathbb{E}_{a\sim\pi_{h+1}(\cdot|s)}\left[\widehat{Q}_{h+1}^{r,\pi}(s,a)\right].$$

This implies that

$$\text{Clip}_{[-(H-h+1),H-h+1]}((w_h^{r,\pi})^\top\phi_h(s,a) + r_h(s,a)) + b_h(s,a) \ge \sum_{s'\in\mathcal{S}} P_h^*(s'|s,a)V_{h+1}^{r,\pi}(s') + r_h(s,a) = Q_h^{r,\pi}(s,a).$$

On the other hand we have

$$Q_h^{r,\pi}(s,a) \le H - h + 1.$$

Therefore we have

$$Q_h^{r,\pi}(s,a) \le \widehat{Q}_h^{r,\pi}(s,a).$$

By induction we can prove the lemma.

### E.4 PROOF OF LEMMA 10

For any two policies $\pi'$ and $\pi$, it follows from the definition of $V^{r,\pi'}$ and $V^{r,\pi}$ that

$$\begin{aligned}
&V^{r,\pi'} - V^{r,\pi} \\
&= \mathbb{E}_{\pi',P^*}\left[r_1(s_1,a_1) + V_2^{r,\pi'}(s_2)\right] - \mathbb{E}_{\pi',P^*}\left[V_1^{r,\pi}(s_1)\right] \\
&= \mathbb{E}_{\pi',P^*}\left[V_2^{r,\pi'}(s_2) - (V_1^{r,\pi}(s_1) - r_1(s_1,a_1))\right] \\
&= \mathbb{E}_{\pi',P^*}\left[V_2^{r,\pi'}(s_2) - V_2^{r,\pi}(s_2)\right] + \mathbb{E}_{\pi',P^*}\left[Q_1^{r,\pi}(s_1,a_1) - V_1^{r,\pi}(s_1)\right] \\
&= \mathbb{E}_{\pi',P^*}\left[V_2^{r,\pi'}(s_2) - V_2^{r,\pi}(s_2)\right] + \mathbb{E}_{\pi',P^*}\left[\langle Q_1^{r,\pi}(s_1,\cdot), \pi_1'(\cdot|s_1) - \pi_1(\cdot|s_1)\rangle\right] \\
&= \cdots = \sum_{h=1}^H \mathbb{E}_{\pi',P^*}\left[\langle Q_h^{r,\pi}(s_h,\cdot), \pi_h'(\cdot|s) - \pi_h(\cdot|s)\rangle\right].
\end{aligned}$$

This concludes our proof.

### E.5 PROOF OF LEMMA 11

For any $h' \in [H]$, suppose $\max_{s\in\mathcal{S}} |\widehat{V}_{h'+1}^{r,\pi}(s) - \widehat{V}_{h'+1}^{r,\pi'}(s)| \le \epsilon_{h'+1}$, then for any $s \in \mathcal{S}, a \in \mathcal{A}$, we have

$$\begin{aligned}
|\widehat{Q}_{h'}^{r,\pi}(s,a) - \widehat{Q}_{h'}^{r,\pi'}(s,a)| &\le |(w_{h'}^{r,\pi} - w_{h'}^{r,\pi'})^\top\phi_{h'}(s,a)| \\
&\le \epsilon_{h'+1}\sum_{i=1}^K |\phi_{h'}(s,a)^\top(\Lambda_{h'})^{-1}\phi_{h'}(s_{h'}^i, a_{h'}^i)| \\
&\le \epsilon_{h'+1}\sqrt{\left[\sum_{i=1}^K \|\phi_{h'}(s,a)\|_{(\Lambda_{h'})^{-1}}^2\right] \cdot \left[\sum_{i=1}^K \|\phi_{h'}(s_{h'}^i, a_{h'}^i)\|_{(\Lambda_{h'})^{-1}}^2\right]} \le \epsilon_{h'+1}\sqrt{dK}.
\end{aligned}$$

Here the final step is comes from the auxiliary Lemma 14 and the fact that $\Lambda_{h'} \ge R^2 I$ and thus $\sum_{i=1}^K \|\phi_{h'}(s,a)\|_{(\Lambda_{h'})^{-1}}^2 \le \sum_{i=1}^K 1 \le K$.

Therefore we have

$$\epsilon_{h'} := \max_{s\in\mathcal{S}} |\widehat{V}_{h'}^{r,\pi}(s) - \widehat{V}_{h'}^{r,\pi'}(s)| \le H\epsilon' + \sqrt{dK}\epsilon_{h'+1}.$$

Note that $\epsilon_{H+1} = 0$, therefore we have

$$\epsilon_1 \le \frac{H\epsilon'}{\sqrt{dK} - 1} \cdot (dK)^{\frac{H}{2}},$$

This concludes our proof.

### E.6 PROOF OF LEMMA 13

The proof is almost the same as Jin et al. (2020b)[Lemma B.3] except that the function class of $V_h^{r,\pi}$ is different. Therefore we only need to bound the covering number $\mathcal{N}_{\mathcal{V}}(\epsilon)$ of $V_h^{r,\pi}$ where the distance is defined as $\mathrm{dist}(V, V') = \sup_s |V(s) - V'(s)|$. Note that $V_h^{r,\pi}$ belongs to the following function class:

$$
\mathcal{V} = \Big\{ V_{w,A}(s) = \mathbb{E}_{a \sim \pi(\cdot|s)} \Big[ \mathrm{Clip}_{[-(H-h+1),H-h+1]}\Big( \mathrm{Clip}_{[-(H-h+1),H-h+1]}(w^\top \phi_{h'}(s,a)) \\
+ \mathrm{Clip}_{[0,2(H-h+1)]}(\|\phi(s,a)\|_A) \Big) \Big], \forall s \in \mathcal{S} \Big\},
$$

where the parameters $(w, A)$ satisfy $\|w\| \le 2H\sqrt{dK/\lambda_{\mathrm{pl}}}, \|A\| \le \beta_{\mathrm{pl}}^2 \lambda_{\mathrm{pl}}^{-1}$.

Note that for any $V_{w_1,A_1}, V_{w_2,A_2} \in \mathcal{V}$, we have

$$
\mathrm{dist}(V_{w_1,A_1}, V_{w_2,A_2}) \le \sup_{s,a} \Big| \Big[ \mathrm{Clip}_{[-(H-h+1),H-h+1]}(w_1^\top \phi_{h'}(s,a)) + \mathrm{Clip}_{[0,2(H-h+1)]}(\|\phi(s,a)\|_{A_1}) \Big]
$$
$$
- \Big[ \mathrm{Clip}_{[-(H-h+1),H-h+1]}(w_2^\top \phi_{h'}(s,a)) + \mathrm{Clip}_{[0,2(H-h+1)]}(\|\phi(s,a)\|_{A_2}) \Big] \Big|
$$
$$
\le \sup_{s,a} \Big| \mathrm{Clip}_{[-(H-h+1),H-h+1]}(w_1^\top \phi_{h'}(s,a)) - \mathrm{Clip}_{[-(H-h+1),H-h+1]}(w_2^\top \phi_{h'}(s,a)) \Big|
$$
$$
+ \sup_{s,a} \Big| \mathrm{Clip}_{[0,2(H-h+1)]}(\|\phi(s,a)\|_{A_1}) - \mathrm{Clip}_{[0,2(H-h+1)]}(\|\phi(s,a)\|_{A_2}) \Big|
$$
$$
\le R \sup_{\|\phi\| \le 1} \Big| (w_1 - w_2)^\top \phi \Big| + R \sup_{\|\phi\| \le 1} \sqrt{\Big| \phi^\top (A_1 - A_2)\phi \Big|}
$$
$$
\le R(\|w_1 - w_2\| + \sqrt{\|A_1 - A_2\|_F}),
$$

where the first and third step utilize the contraction property of Clip. Let $\mathcal{C}_w$ be the $\epsilon/(2R)$-cover of $\{w \in \mathbb{R}^d : \|w\| \le 2r_{\max}\sqrt{dK/\lambda_{\mathrm{pl}}}\}$ w.r.t. $\ell_2$-norm and $\mathcal{C}_A$ be the $(\epsilon/2R)$-cover of $\{A \in \mathbb{R}^{d \times d} : \|A\| \le \beta_{\mathrm{pl}}^2 \lambda_{\mathrm{pl}}^{-1}\}$ w.r.t. the Frobenius norm, then from the literature Jin et al. (2020b)[Lemma D.5], we have

$$
\mathcal{N}_{\mathcal{V}}(\epsilon) \le \log|\mathcal{C}_w| + \log|\mathcal{C}_A| \le d\log\Big(1 + 8\sqrt{dKr_{\max}^2 R^2/(\lambda_{\mathrm{pl}}\epsilon^2)}\Big) + d^2 \log\Big[1 + 8d^{1/2}\beta_{\mathrm{pl}}^2 R^2/(\lambda_{\mathrm{pl}}\epsilon^2)\Big].
\tag{24}
$$

The rest of the proof follows Jin et al. (2020b)[Lemma B.3] directly so we omit it here.

### E.7 PROOF OF PROPOSITION 1

First consider $\zeta$ and $\zeta'$ which satisfies:

$$
\|\zeta_h - \zeta_h'\| \le \epsilon_z, \forall h \in [H].
$$

Then we know for any $h \in [H], s \in \mathcal{S}, a \in \mathcal{A}$,

$$
|\zeta_h^\top \phi_h(s,a) - (\zeta_h')^\top \phi_h(s,a)| \le \epsilon_z R.
\tag{25}
$$

Now fix any $h \in [H]$ and $s \in \mathcal{S}$. To simplify writing, we use $x(a)$ and $x'(a)$ to denote $\zeta_h^\top \phi_h(s,a)$ and $(\zeta_h')^\top \phi_h(s,a)$ respectively. Without loss of generality, we assume $\sum_a \exp(x(a)) \le \sum_a \exp(x'(a))$. Then from (25) we have

$$
\sum_a \exp(x(a)) \le \sum_a \exp(x'(a)) \le \exp(\epsilon_z R) \sum_a \exp(x(a)).
$$

Note that we have

$$
\|\pi_h^\zeta(\cdot|s) - \pi_h^{\zeta'}(\cdot|s)\|_1 = \sum_a \Big| \frac{\exp(x(a))}{\sum_{a'} \exp(x(a'))} - \frac{\exp(x'(a))}{\sum_{a'} \exp(x'(a'))} \Big|
$$

$$= \frac{\sum_a \left| \exp(x(a)) \sum_{a'} \exp(x'(a')) - \exp(x'(a)) \sum_{a'} \exp(x(a')) \right|}{\sum_{a'} \exp(x(a')) \cdot \sum_{a'} \exp(x'(a'))}.$$

For any $a \in \mathcal{A}$, if $\exp(x(a)) \sum_{a'} \exp(x'(a')) - \exp(x'(a)) \sum_{a'} \exp(x(a')) \geq 0$, then

$$\left| \exp(x(a)) \sum_{a'} \exp(x'(a')) - \exp(x'(a)) \sum_{a'} \exp(x(a')) \right|$$

$$\leq \exp(\epsilon_z R) \exp(x(a)) \sum_{a'} \exp(x(a')) - \exp(-\epsilon_z R) \exp(x(a)) \sum_{a'} \exp(x(a'))$$

$$= (\exp(\epsilon_z R) - \exp(-\epsilon_z R)) \exp(x(a)) \sum_{a'} \exp(x(a')).$$

Otherwise, we have

$$\left| \exp(x(a)) \sum_{a'} \exp(x'(a')) - \exp(x'(a)) \sum_{a'} \exp(x(a')) \right|$$

$$\leq \exp(\epsilon_z R) \exp(x(a)) \sum_{a'} \exp(x(a')) - \exp(x(a)) \sum_{a'} \exp(x(a'))$$

$$= (\exp(\epsilon_z R) - 1) \exp(x(a)) \sum_{a'} \exp(x(a')).$$

Therefore we have

$$\|\pi_h^\zeta(\cdot|s) - \pi_h^{\zeta'}(\cdot|s)\|_1 \leq \frac{(\exp(\epsilon_z R) - \exp(-\epsilon_z R)) \sum_a \exp(x(a)) \sum_{a'} \exp(x(a'))}{\sum_{a'} \exp(x(a')) \cdot \sum_{a'} \exp(x'(a'))} \leq \exp(2\epsilon_z R) - 1.$$

This implies that for any $\epsilon \leq 1$,

$$\mathcal{N}_\Pi(\epsilon) \leq \left( \mathcal{N}_{\mathbb{B}(d,W)} \left( \frac{\ln 2}{2R} \epsilon \right) \right)^H \leq \left( \frac{12WR}{\epsilon} \right)^{Hd},$$

where the first step uses $\exp(x) - 1 \leq x/\ln 2$ when $x \leq \ln 2$. This concludes our proof.

### E.8 PROOF OF PROPOSITION 2

First we consider the following entropy-regularized RL problem where we try to maximize the following objective for some $\alpha > 0$:

$$\max_\pi V_\alpha(r^*, \pi) := \mathbb{E}_{\pi, P*} \left[ \sum_{h=1}^H r_h^*(s_h, a_h) - \alpha \log \pi_h(a_h|s_h) \right].$$

From the literature (Nachum et al., 2017; Cen et al., 2022), we know that we can define corresponding optimal regularized value function and Q function as follows:

$$Q_{\alpha,h}^*(s, a) = r_h^*(s, a) + \mathbb{E}_{s_{h+1} \sim P_h^*(\cdot|s,a)} \left[ V_{\alpha,h+1}^* \right],$$

$$V_{\alpha,h}^*(s) = \max_{\pi_h} \mathbb{E}_{a_h \sim \pi_h(\cdot|s)} \left[ Q_{\alpha,h}^*(s, a_h) - \alpha \log \pi_h(a_h|s) \right],$$

where $V_{\alpha,H+1}^*(s) = 0$ for all $s \in \mathcal{S}$. Note that we have $V_{\alpha,h}^*(s) \leq H(1 + \alpha \log |\mathcal{A}|)$ for all $s \in \mathcal{S}$ and $h \in [H]$. The global optimal regularized policy is therefore

$$\pi_{\alpha,h}^*(a|s) = \frac{\exp(Q_{\alpha,h}^*(s, a)/\alpha)}{\sum_{a'} \exp(Q_{\alpha,h}^*(s, a')/\alpha)}.$$

In particular, in linear MDPs, we have

$$Q_{\alpha,h}^*(s, a) = \phi_h(s, a)^\top \left( \theta_h^* + \int_{s \in \mathcal{S}} \mu_h^*(s) V_{\alpha,h+1}^*(s) ds \right).$$

Therefore, $Q_{\alpha,h}^*(s,a) = \phi_h(s,a)^\top w_{\alpha,h}^*$ where

$$\|w_{\alpha,h}^*\| \le B + H(1 + \alpha \log|\mathcal{A}|)\sqrt{d}.$$

This implies that $\pi_\alpha^*$ belongs to the log-linear policy class $\Pi$ with $W = (B + H(1 + \alpha \log|\mathcal{A}|)\sqrt{d})/\alpha$.

On the other hand, let $\pi_g$ denote the global unregularized optimal policy, then

$$V^*(r^*, \pi_g) - \max_{\pi \in \Pi} V(r^*, \pi) \le V^*(r^*, \pi_g) - V(r^*, \pi_\alpha^*)$$

$$= \left(V^*(r^*, \pi_g) - V_\alpha^*(r^*, \pi_g)\right) + \left(V_\alpha^*(r^*, \pi_g) - V_\alpha^*(r^*, \pi_\alpha^*)\right) + \left(V_\alpha^*(r^*, \pi_\alpha^*) - V^*(r^*, \pi_\alpha^*)\right)$$

$$\le V_\alpha^*(r^*, \pi_\alpha^*) - V^*(r^*, \pi_\alpha^*) \le \alpha H \log|\mathcal{A}|.$$

Therefore we only need to let $\alpha = \frac{\epsilon}{H \log|\mathcal{A}|}$ to ensure $V(r^*, \pi_g) - \max_{\pi \in \Pi} V(r^*, \pi) \le \epsilon$.

## F  PROOF OF THEOREM 3

First from performance difference lemma (Lemma 10), we have

$$V^{r^*, \widehat{\pi}} - V^{r^*, *} = \sum_{h=1}^H \mathbb{E}_{s_h \sim d_h^{\widehat{\pi}}}[Q_h^*(s_h, \widehat{\pi}) - Q_h^*(s_h, \pi^*)]$$

$$= \sum_{h=1}^H \mathbb{E}_{s_h \sim d_h^{\widehat{\pi}}}[Q_h^*(s_h, \widehat{\pi}) - \widehat{A}_h(s_h, \widehat{\pi})] + \mathbb{E}_{s_h \sim d_h^{\widehat{\pi}}}[\widehat{A}_h(s_h, \widehat{\pi}) - \widehat{A}_h(s_h, \pi^*)]$$

$$+ \mathbb{E}_{s_h \sim d_h^{\widehat{\pi}}}[\widehat{A}_h(s_h, \pi^*) - Q_h^*(s_h, \pi^*)]$$

$$\ge \sum_{h=1}^H \mathbb{E}_{s_h \sim d_h^{\widehat{\pi}}}[Q_h^*(s_h, \widehat{\pi}) - \widehat{A}_h(s_h, \widehat{\pi})] + \mathbb{E}_{s_h \sim d_h^{\widehat{\pi}}}[\widehat{A}_h(s_h, \pi^*) - Q_h^*(s_h, \pi^*)]$$

$$= \sum_{h=1}^H \mathbb{E}_{s_h \sim d_h^{\widehat{\pi}}}[A_h^*(s_h, \widehat{\pi}) - \widehat{A}_h(s_h, \widehat{\pi}) + \widehat{A}_h(s_h, \pi^*) - A_h^*(s_h, \pi^*)]$$

$$= \sum_{h=1}^H \mathbb{E}_{s_h \sim d_h^{\widehat{\pi}}}[\langle \phi_h(s_h, \widehat{\pi}), \xi_h^* - \widehat{\xi}_h \rangle - \langle \phi_h(s_h, \pi^*), \xi_h^* - \widehat{\xi}_h \rangle]$$

$$= \sum_{h=1}^H \mathbb{E}_{s_h \sim d_h^{\widehat{\pi}}}[\langle \phi_h(s_h, \widehat{\pi}) - \phi_h(s_h, \pi^*), \xi_h^* - \widehat{\xi}_h \rangle]$$

$$\ge -\sum_{h=1}^H \|\mathbb{E}_{s_h \sim d_h^{\widehat{\pi}}}[\phi_h(s_h, \widehat{\pi}) - \phi_h(s_h, \pi^*)]\|_{\Sigma_{h,N+1}^{-1}} \cdot \|\xi_h^* - \widehat{\xi}_h\|_{\Sigma_{h,N+1}}. \tag{26}$$

Next we will bound $\|\mathbb{E}_{s_h \sim d_h^{\widehat{\pi}}}[\phi_h(s_h, \widehat{\pi}) - \phi_h(s_h, \pi^*)]\|_{\Sigma_{h,N+1}^{-1}}$ and $\|\xi^* - \widehat{\xi}\|_{\Sigma_{h,N+1}}$ respectively. First for $\|\mathbb{E}_{s_h \sim d_h^{\widehat{\pi}}}[\phi_h(s_h, \widehat{\pi}) - \phi_h(s_h, \pi^*)]\|_{\Sigma_{h,N+1}^{-1}}$, notice that $\Sigma_{h,N+1} \succeq \Sigma_{h,n}$ for all $n \in [N+1]$, which implies

$$\|\mathbb{E}_{s_h \sim d_h^{\widehat{\pi}}}[\phi_h(s_h, \widehat{\pi}) - \phi_h(s_h, \pi^*)]\|_{\Sigma_{h,N+1}^{-1}} \le \frac{1}{N} \sum_{n=1}^N \|\mathbb{E}_{s_h \sim d_h^{\widehat{\pi}}}[\phi_h(s_h, \widehat{\pi}) - \phi_h(s_h, \pi^*)]\|_{\Sigma_{h,n}^{-1}}$$

$$\le \frac{1}{N} \sum_{n=1}^N \|\mathbb{E}_{s_h \sim \pi^{h,n,0}}[\phi_h(s_h, \pi^{h,n,0}) - \phi_h(s_h, \pi^{h,n,1})]\|_{\Sigma_{h,n}^{-1}}$$

$$\le \frac{1}{\sqrt{N}} \sqrt{\sum_{n=1}^N \|\mathbb{E}_{s_h \sim \pi^{h,n,0}}[\phi_h(s_h, \pi^{h,n,0}) - \phi_h(s_h, \pi^{h,n,1})]\|_{\Sigma_{h,n}^{-1}}^2}$$

$$\le \sqrt{\frac{2d \log(1 + N/d)}{N}}, \tag{27}$$

where the third step comes from the definition of $\pi^{h,n,0}$ and $\pi^{h,n,1}$ and the last step comes from Elliptical Potential Lemma (Lemma 1) and the fact that $\lambda \geq 4R^2$.

For $\|\xi_h^* - \widehat{\xi}_h\|_{\Sigma_{h,N+1}}$, let $\widetilde{\Sigma}_{h,n}$ denote $\lambda I + \sum_{i=1}^{n-1}(\phi_h(s^{h,n}, a^{h,n,0}) - \phi_h(s^{h,n}, a^{h,n,1}))(\phi_h(s^{h,n}, a^{h,n,0}) - \phi_h(s^{h,n}, a^{h,n,1}))^\top$. Then similar to Lemma 3, we have with probability at least $1 - \delta/2$,

$$\|\xi_h^* - \widehat{\xi}_h\|_{\Sigma_{h,N+1}}^2 \leq 2\|\xi_h^* - \widehat{\xi}_h\|_{\widetilde{\Sigma}_{h,N+1}}^2 + 2C_{\mathrm{CON}}dR^2B^2\log(N/\delta). \tag{28}$$

On the other hand, similar to Lemma 2, MLE guarantees us that with probability at least $1 - \delta/2$,

$$\|\widehat{\xi}_h - \xi_h^*\|_{\widetilde{\Sigma}_{h,N+1}} \leq 2C_{\mathrm{MLE}} \cdot \sqrt{\kappa_{\mathrm{adv}}^2(d + \log(1/\delta)) + \lambda B^2}, \tag{29}$$

where $\kappa_{\mathrm{adv}} = 2 + \exp(2B_{\mathrm{adv}}) + \exp(-2B_{\mathrm{adv}})$.

Therefore combining (28) and (29), we have with probability at least $1 - \delta$,

$$\|\xi^* - \widehat{\xi}_h\|_{\Sigma_{h,N+1}} \leq \mathcal{O}(\kappa_{\mathrm{adv}}BR\sqrt{\lambda d\log(N/\delta)}). \tag{30}$$

Thus combining (26), (27) and (30) via union bound, we have $V^*(r^*) - V(r^*, \widehat{\pi}) \leq \epsilon$ with probability at least $1 - \delta$ as long as

$$N \geq \widetilde{\mathcal{O}}\Big(\frac{\lambda\kappa_{\mathrm{adv}}^2 B^2 R^2 H^2 d^2 \log(1/\delta)}{\epsilon^2}\Big).$$

## G   AUXILIARY LEMMAS

**Lemma 14** (Jin et al. (2020b)[Lemma D.1]). *Let $\Lambda = \lambda I + \sum_{i=1}^K \phi_i\phi_i^\top$ where $\phi_i \in \mathbb{R}^d$ and $\lambda > 0$, then we have $\sum_{i=1}^K \phi_i^\top \Lambda^{-1} \phi_i \leq d$.*

