---

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

'})^{-1}}\Big] \cdot \Big[\sum_{i=1}^{K} \|\phi_{h'}(s^i_{h'},a^i_{h'})\|^2_{(\Lambda_{h'})^{-1}}\Big]} \leq \epsilon_{h'+1}\sqrt{dK}.$$

Here the final step is comes from the auxiliary Lemma 14 and the fact that $\Lambda_{h'} \geq R^2 I$ and thus $\sum_{i=1}^{K} \|\phi_{h'}(s,a)\|^2_{(\Lambda_{h'})^{-1}} \leq \sum_{i=1}^{K} 1 \leq K$.

Therefore we have

$$\epsilon_{h'} := \max_{s \in \mathcal{S}} |\widehat{V}^{r,\pi}_{h'}(s) - \widehat{V}^{r,\pi'}_{h'}(s)| \leq H\epsilon' + \sqrt{dK}\epsilon_{h'+1}.$$

Note that $\epsilon_{H+1} = 0$, therefore we have

$$\epsilon_1 \leq \frac{H\epsilon'}{\sqrt{dK}-1} \cdot (dK)^{\frac{H}{2}},$$

This concludes our proof.

### E.6 PROOF OF LEMMA 13

The proof is almost the same as Jin et al. (2020b)[Lemma B.3] except that the function class of $V^{r,\pi}_h$ is different. Therefore we only need to bound the covering number $\mathcal{N}_{\mathcal{V}}(\epsilon)$ of $V^{r,\pi}_h$ where the distance is defined as $\text{dist}(V,V') = \sup_s |V(s) - V'(s)|$. Note that $V^{r,\pi}_h$ belongs to the following function class:

$$\mathcal{V} = \Bigg\{ V_{w,A}(s) = \mathbb{E}_{a\sim\pi(\cdot|s)}\Bigg[\text{Clip}_{[-(H-h+1),H-h+1]}\Bigg(\text{Clip}_{[-(H-h+1),H-h+1]}(w^\top \phi_{h'}(s,a))$$
$$+\text{Clip}_{[0,2(H-h+1)]}(\|\phi(s,a)\|_A)\Bigg)\Bigg], \forall s \in \mathcal{S}\Bigg\},$$

where the parameters $(w,A)$ satisfy $\|w\| \leq 2H\sqrt{dK/\lambda_{\text{pl}}}, \|A\| \leq \beta^2_{\text{pl}}\lambda^{-1}_{\text{