# OpenReview forum: "Provable Reward-Agnostic Preference-Based Reinforcement Learning"
_ICLR.cc/2024/Conference — ICLR 2024 spotlight_

### Official Review · Reviewer_XEZ8 · 2023-10-29

**Soundness:** 3 good
**Presentation:** 3 good
**Contribution:** 3 good
**Rating:** 8
**Confidence:** 4

**Summary:**

This paper deals with Preference-based RL (PbRL), a framework in which the learning agent collects a pair of trajectories and asks an external evaluator which one is better. Differently from previous works, this paper proposes to decouple the exploration of the environment, which is done through a reward-free oracle, and the active learning of the reward, by querying the external evaluators on previously collected trajectory pairs. Specifically, the paper provides (i) new algorithms implementing this framework for PbRL in both tabular and linear MDPs, which are called REGIME and REGIME-lin respectively, (ii) the analysis of the sample complexity of the latter algorithms, (iii) an alternative feedback model in which the external evaluator provides preferences over action pairs, together with a corresponding algorithm variation that is called REGIME-action.

**Strengths:**

- (Originality) To the best of my knowledge, this is the first paper providing sample complexity results for PbRL with an algorithm specifically designed for the PAC setting instead of regret minimization;
- (Significance) As the authors argue in the paper, their framework where exploration and feedback collection are decoupled can suit better some important applications;
- (Quality) Although I did not checked the derivations in details, the reported results and the considered assumptions look reasonable;
- (Clarity) The paper is well presented and clear, even if some comparison with previous works seem overstretched.

**Weaknesses:**

- (Worst-case complexity) All of the reported results on trajectory-preference feedback come with the huge caveat that the rate may be exponential in the worst-case, as $r_{\text{max}}$ can be as large as $H$;
- (Lower bound) The paper does not provide a lower bound on the sample complexity, which makes unclear whether the reported factors are actually unavoidable (especially in $\kappa$ as the necessity of $|\mathcal{S}|^2 |\mathcal{A}|$ cannot be overcome due to the reward-free oracle).

GENERAL COMMENT

This looks like an interesting paper addressing a relevant problem, which is gaining additional traction given the recent success of RL from human feedback. I found the claim of improved results w.r.t. prior works (e.g., Pacchiano et al., 2021) to be a little overstated: Previous works target regret minimization instead of sample complexity, which arguably makes the comparison spurious. Whereas it is important to show that REGIME improves their sample complexity rate, it is unclear whether the regret can also be bounded, which would result in a worse fit for online settings. Having said that, both the online settings and the online exploration/offline feedback collection look reasonable to me, hence the results in this paper are valuable. For this reason, I am currently providing a slightly positive evaluation, but I am open to increase my score after a deeper inspection of the analysis possibly revealing interesting techniques, and a convincing authors' response on the questions below.

**Questions:**

1) While it is widely known that Markovian policies are sufficient to maximize a reward function, it comes as a surprise history-dependent policies are not considered although the feedback is based on full trajectories. I guess this might be a consequence of the linear reward assumption. Can the authors develop on why Markovian policies are sufficient to actively generate trajectories for the feedback collection step?

2) In the Preliminaries section, the paper claims that "it is necessary to make structural assumptions about the reward". Do the authors mean that the problem is not learnable without the linear reward assumption, or that *some* structural assumption is needed? Providing a lower bound would be clarifying of course.

3) Related to the previous question, one may wonder whether assuming an underlying reward function is really necessary for PbRL. Can the problem be cast as a direct maximization of the human feedback instead?

4) The action-based preference feedback looks somehow related to inverse RL, where the learner can query an expert on the optimal action for a given state rather than a preference between a pair of actions. Can the authors relate their findings with previous inverse RL literature? Do they think their problem is easier/harder than inverse RL?

5) Can the authors discuss the novelty of their sample complexity analysis w.r.t. prior work? Is there any novel technique they would like to highlight?

---

> ### Author Response · Authors · 2023-11-16
> **Response (Part 1)**
>
> Thank you for the valuable feedback! Here are our responses.
>
> - **Worst-case Sample Complexity (in Weaknesses)**
>
> We admit that $\kappa$ will scale with $\exp(H)$ in the worst case. This quantity occurs in **all the existing theoretical works** on PbRL [1-3] and seems inevitable for MLE estimators. That said, in the **sparse-reward setting** which is very common in practice, $r _ {max}$ is typically $O(1)$ and thus $\kappa$ will also be negligible. In addition, to circumvent the scaling of $\kappa$, we also consider **action-based setting** in the paper where we only scale with $\exp(B _ {adv})$, which is guaranteed to be not greater than $r _ {max}$ and often much smaller in practice [4].
>
> - **General comments (in Weaknesses)**
>
> We agree that [1] focuses on regret minimization while our goal is to improve sample complexities in terms of PAC-learnability. We will make it more clear in our final version.
>
> - **Why Markovian Policies Are Sufficient**
>
> This is an excellent point; but, this is not due to linear reward parametrization. **Instead, it is because while human feedback relies on the whole trajectories, rewards in actual MDPs are per-step rewards**, i.e., the reward function is a mapping from $\mathcal{S}\times\mathcal{A}\times [H]$ to $[0,1]$. Hence, in this MDP,  there exists **an optimal policy which is Markovian** [5]. Therefore, in Step 4 we only need to search within Markovian policies too. Note that the objective of Step 1 is to ensure the dataset can cover the optimal policy and arbitrary output policies such that the estimated reward is accurate under these policies. Now that the optimal policy and output policy are both Markovian, we only need to consider Markovian policies in lines 4-7 in Algorithm 1.
>
> On the other hand, If we consider more general trajectory-wise reward, i.e., the reward function maps directly from a trajectory to $[0, r _ {max}]$ as in other PbML work [2], the optimal policy might not be Markovian and then we will also need to consider non-Markovian policies in line 4-7 in Algorithm 1.
>
> - **Requirement of Linear Reward Parametrization**
>
> We agree that “it is necessary to make structural assumptions about the reward” is a bit informal. We were not trying to make a formal statement here. We will change the wording into a “we make structural assumptions as in other works [1,3]”.
>
> In a nutshell, here originally, we just wanted to convey that **we need some structural assumptions for generalization, and we chose linear models as these structural assumptions**. For example, in the literature on standard supervised learning, the former statement (necessary to assume structure) is often formalized as VC dimension needs to be finite to ensure learnability.  Yeah, but to say formally in our context, we need to establish certain lower bounds as you mention. Hence, we will just remove it.
>
> - **Direct Maximization**
>
> This is a good point. Direct maximization of human preference is possible and has been studied theoretically as the well-known dueling bandits problem [8-10]. However, in practice, in PbRL, people usually first train a reward model and then use the reward model for downstream learning [11-15]. We think this is because after learning a reward model, people can use the reward model to score an arbitrary number of trajectories or even reuse the model in other related tasks, thus reducing a large amount of human cost. Therefore, due to the wide application of such a framework, we also assume an underlying reward model in this paper.
>
> - **Relation to Inverse RL**
>
> While there are some similarities as you mention, we think **PbRL is still very different from inverse RL**. Most importantly, in inverse RL, the actions are sampled from an expert policy directly and thus we only need to learn a reward function that matches such expert behaviors. However, in PbRL, the pair of actions is not sampled from an expert and can possibly be undesirable.

---

> ### Author Response · Authors · 2023-11-16
> **Response (Part 2)**
>
> - **Technical Novelty**
>
> The novelty of our techniques mainly lies in **the design and analysis of the reward-agnostic trajectory collection procedure (line 4-7 in Algorithm 1)**.  We will try to emphasize more in the final version. More specifically, our analysis deals with the following challenges that are not solved in the literature:
>
> 1. **First, almost all the reward-free exploration literature is focused on learning the transition dynamics of the MDPs. Our problem is completely different because our main task is to learn the reward function from human preferences**. Consequently, our iterative collection procedure (line 4-7 in Algorithm 1) also differs from the reward-free exploration literature. For example, [17] is the closest work to ours and studies the reward-free exploration in linear MDPs. They utilize a variant of UCB-LSVI, which is quite different from our collection procedure. Correspondingly, we can not directly use the existing analysis either and need to come up with new proofs.
>
> 2. **Second, our framework is also different from the existing online PbRL [6-7]**. In their algorithms, the reward-learning process and exploration phase are not decoupled and the analysis heavily relies on the utilization of **optimism**. In contrast, our algorithm **does not use optimism**, and thus their analysis techniques also do not apply here.
>
> 3. **Third, our analysis of  REGIME-lin differs from the one in the literature on linear MDPs**. This is because while the style of our algorithm is akin to **multi-policy evaluation** in linear MDPs, the current literature of linear MDPs mainly focuses on **Q-learning-style algorithms for policy optimization** [16-17].
>
> More specifically, in REGIME-lin, we want to **evaluate all the policies in a policy class under a certain reward function**. This brings a new challenge that does not occur in [16-17]: our learning difficulty will scale with the size of the policy class $\Pi$. This forces us to find and analyze **an appropriate policy class that is able to approximate the optimal policy while retaining a relatively small size**. In addition, a key component of our analysis is the **new concentration inequality** in Lemma 13, which also differs from the existing concentration lemma in [16-17] and may be of independent interest.

---

> > ### Comment · Reviewer_XEZ8 · 2023-11-17
> > **After Response**
> >
> > I want to thank the authors for their detailed replies, which covered all of my questions, and for explaining the technical novelty in detail. I would suggest to include the latter in a final version of the paper. I am increasing my score accordingly.
> >
> > **Inverse RL**
> >
> > While I was not implying that the proposed setting with action feedback is exactly an inverse RL setting, the connection might be stricter than what the authors believe. Especially, if one fixes a state and queries the preference for two actions, then queries the best of the two with another action, and so on... I guess the "expert" action can be quickly discovered (as long as the action set is finite). When you have got the "expert" action this arguably reduces to IRL. Thus, I was wondering whether the reward identification rate can be related to IRL rates (e.g., Lindner et al., Active Exploration for Inverse Reinforcement Learning, 2022; Metelli et al., Towards Theoretical Understanding of Inverse Reinforcement Learning, 2023). There might be other crucial differences that I am not considering, but the comparison may worth a discussion in the paper.

---

> > > ### Author Response · Authors · 2023-11-20
> > >
> > > Thank you so much for increasing the score, and we will include the technical novelty in the final version.
> > >
> > > In relation to inverse RL, we agree that the reduction that the author proposed could work. However, we believe that the identification of the expert action is more complicated. Since the human preference is random, we need to sample multiple human feedbacks for one pair of actions to identify the better action accurately. From standard concentration inequality, the number of human feedback we require for each pair of actions should scale with the inverse of the reward difference of the two actions. Consequently, when the reward difference is very small, we might incur a large number of human feedback just to identify which one is better between these two actions. Therefore, it can be quite hard and expensive to find the exact expert action in our setting.

---

> > > > ### Comment · Reviewer_XEZ8 · 2023-11-22
> > > >
> > > > Thank you for the follow-up comment.
> > > >
> > > > Perhaps when the reward difference is very small, identifying the expert action does not matter much if the goal is to find a nearly optimal policy. Anyway, it is clear that this is not trivial and it is perfectly fine if the paper does not address that.

---

> > > > > ### Author Response · Authors · 2023-11-22
> > > > >
> > > > > You are welcome!

---

> ### Author Response · Authors · 2023-11-16
> **Response (Part 3)**
>
> [1] Saha, Aadirupa, Aldo Pacchiano, and Jonathan Lee. "Dueling RL: Reinforcement Learning with Trajectory Preferences." International Conference on Artificial Intelligence and Statistics. PMLR, 2023.
>
> [2] Chen, Xiaoyu, et al. "Human-in-the-loop: Provably efficient preference-based reinforcement learning with general function approximation." International Conference on Machine Learning. PMLR, 2022.
>
> [3] Zhu, Banghua, Jiantao Jiao, and Michael I. Jordan. "Principled Reinforcement Learning with Human Feedback from Pairwise or $ K $-wise Comparisons." arXiv preprint arXiv:2301.11270 (2023).
>
> [4] Ross, Stéphane, Geoffrey Gordon, and Drew Bagnell. "A reduction of imitation learning and structured prediction to no-regret online learning." Proceedings of the fourteenth international conference on artificial intelligence and statistics. JMLR Workshop and Conference Proceedings, 2011.
>
> [5] Bertsekas, Dimitri. Dynamic programming and optimal control: Volume I. Vol. 4. Athena scientific, 2012.
>
> [6] Jin, Chi, Qinghua Liu, and Sobhan Miryoosefi. "Bellman eluder dimension: New rich classes of rl problems, and sample-efficient algorithms." Advances in neural information processing systems 34 (2021): 13406-13418.
>
> [7] Huang, Baihe, et al. "Towards general function approximation in zero-sum markov games." arXiv preprint arXiv:2107.14702 (2021).
>
> [8] Yue, Yisong, et al. "The k-armed dueling bandits problem." Journal of Computer and System Sciences 78.5 (2012): 1538-1556.
>
> [9] Zoghi, Masrour, et al. "Relative confidence sampling for efficient on-line ranker evaluation." Proceedings of the 7th ACM international conference on Web search and data mining. 2014.
>
> [10] Dudík, Miroslav, et al. "Contextual dueling bandits." Conference on Learning Theory. PMLR, 2015.
>
> [11] Ouyang, Long, et al. "Training language models to follow instructions with human feedback." Advances in Neural Information Processing Systems 35 (2022): 27730-27744.
>
> [12] Glaese, Amelia, et al. "Improving alignment of dialogue agents via targeted human judgements." arXiv preprint arXiv:2209.14375 (2022).
>
> [13] Ramamurthy, Rajkumar, et al. "Is reinforcement learning (not) for natural language processing?: Benchmarks, baselines, and building blocks for natural language policy optimization." arXiv preprint arXiv:2210.01241 (2022).
>
> [14] Xue, Wanqi, et al. "PrefRec: Preference-based Recommender Systems for Reinforcing Long-term User Engagement." arXiv preprint arXiv:2212.02779 (2022).
>
> [15] Shin, Daniel, Anca D. Dragan, and Daniel S. Brown. "Benchmarks and algorithms for offline preference-based reward learning." arXiv preprint arXiv:2301.01392 (2023).
>
> [16] Jin, Chi, et al. "Provably efficient reinforcement learning with linear function approximation." Conference on Learning Theory. PMLR, 2020.
>
> [17] Wang, Ruosong, et al. "On reward-free reinforcement learning with linear function approximation." Advances in neural information processing systems 33 (2020): 17816-17826.

---

### Official Review · Reviewer_dgpw · 2023-10-30

**Soundness:** 3 good
**Presentation:** 3 good
**Contribution:** 3 good
**Rating:** 8
**Confidence:** 2

**Summary:**

The authors introduce an algorithm for PbRL, that collects trajectories and preference feedback only in an initial phase and not iteratively. Assuming a linear reward function and an epsilon-exact transition function estimator, they show a novel sample-complexity analysis. This approach is also extended to a linear MDP setting, where it is possible to forgo the requirement of an transition function estimator and directly approximate a policies feature space. They also show a complexity analysis that is independent of $r_{max}$.

**Strengths:**

The authors consider a relevant problem, which relates to a commonly used variant of PbRL (non-interactive). The given assumptions are reasonable under for many real world scenarios. Linear rewards can usually be achieved with suitable projections and epsilon-exact transition function approximators are also available in several domains. Therefore, the complexity analysis is potentially applicable to a wide range of problems, rendering the work significant. The work is also original, as the authors deviate from the common, reward-based scheme.
Clarity could be slightly improved, as explained in the following.

**Weaknesses:**

The most substantial weakness of the contribution, is its substantial dependence on the appendix. It is acceptable to move specific details, like a formal proof, to the supplementary, but the main paper should be able to stand on its own. This mostly concerns Theorem 1, which is not sufficiently explained in the main paper. At least the basic idea/concept should be added. Furthermore, Algorithm 4/5 are direct references to the supplementary material.

On a side note, the relation to $\hat{P}$ and the related approximation error is not obvious from the used notation, because the dependence is missing from line 5 (and14) of Algorithm 1.

**Questions:**

-

---

> ### Author Response · Authors · 2023-11-16
> **Response**
>
> Thank you for the positive feedback! Here are our responses.
>
> - **Presentation**
>
> Thanks for the suggestions! We will move some key points in the Appendix to the main paper.
>
> - **Relation to $\widehat{P}$**
>
> We use $\widehat{\phi}$ and $\widehat{\Sigma}$ to denote the quantities evaluated under $\widehat{P}$ to simplify the writing. We will further remark this in the paper.

---

> ### Author Response · Authors · 2023-11-20
>
> Dear reviewer,
>
> There are only three days left for the discussion period. Do you have any other questions that we can help with?

---

> > ### Comment · Reviewer_dgpw · 2023-12-01
> >
> > Dear authors,
> > thanks for the short reply. However, it does not change my evaluation.

---

### Official Review · Reviewer_m5cB · 2023-10-30

**Soundness:** 3 good
**Presentation:** 2 fair
**Contribution:** 3 good
**Rating:** 8
**Confidence:** 2

**Summary:**

This paper studies Preference-based RL (PbRL) for tabular, linear, and low-rank MDPs. The authors focus on designing PAC algorithms and prove sample complexity guarantees for all proposed algorithms. In particular, the proposed algorithms separate the trajectory selection stage and human feedback stage, which could be very useful in applications since all the human feedback can be queried in a single batch. Finally, the authors also extend their algorithms to action-based comparisons.

**Strengths:**

- The studied problem of PAC PbRL is interesting and hasn't been studied much before.
- As far as I can tell, the contributions of this paper are novel and significant in that (a) the algorithms attain better sample complexity and (b) are arguably more practical by seperating the trajectory collection and human feedback stages.
- A main strength of the porposed algorithms is that all the sampling of exploratory trajcetories is done first and only then human comparisons queried.

**Weaknesses:**

- I think that the presentation can be improved. I know that space limitations are tight, however, removing margins around equations and sections will make the paper much harder to read. A table which compares this paper's results with related work would also help the reader to place this work into the existing literature. This would be particularly helpful since much of the related work does not study PAC algorithms, but only regret minimization, and it is difficult to figure out the current state-of-the-art and open questions in this area from just reading the paper.

**Questions:**

- At the end of Section 3, you compare to Pacchiano et al. (2021). I suppose that the stated sample complexities are for $(\varepsilon, \delta)$-PAC and the dependence on $\delta$ omitted? I'm also slightly confused because the referenced Theorem 2 in Pacchiano et al. (2021) only provides a regret upper bound. Where did you find the PAC bound?
- How computationally expensive is step 1 (e.g., line 5 in Algorithm 1, line 7 in Algorithm 2, line 5 in Algorithm 3)?
- How do you think will your algorithms perform in practice? Do you anticipate any obstacles when deploying your algorithms (computational or otherwise)?
- Intuitively, one would think that adaptive trajectory selection which queries human feedback after every selection would be advantageous compared to first selecting all trajectories without observing intermediate human feedback. In other words, you would expect *adaptive* experimental design to perform better than *offline* experimental design (even when it comes to PAC learning). Your results however suggest that "offline" trajectory selection does not hurt performance. Can you provide any reasons or intuition for this?
- Hereto related, do you think that your approach of seperating trajectory selection and human feedback stages is applicable/useful for regret minimization as well? (Beyond trivially reducing your $(\varepsilon, \delta)$-PAC guarantees to regret bounds via some explore-then-commit strategy).

---

> ### Author Response · Authors · 2023-11-16
> **Response (Part 1)**
>
> Thank you for the positive feedback! Here are our responses:
>
> - **Presentation**
>
> Thanks for the suggestions! We will add more margins around equations in the final version. In addition, as far as we know, the existing theoretical online PbRL works [1-2] focus on regret minimization and only an offline PbRL paper [3] studies PAC algorithm. The PAC sample complexity of [1-2] is derived from the corresponding regret bound. We will try to add the table in the final version.
>
> - **Sample Complexity in [1]**
>
> We are sorry that a $\log(1/\delta)$ term is missed in the complexity. We will correct this in the final version. In addition, [1] only studies the regret bound so we derive the PAC sample complexity directly from the regret bound, i.e., $K$ such that $reg (K ) / K \leq \epsilon$. As far as we know, the existing theoretical online PbRL works [1-2] focus on regret minimization and do not have a specific PAC learning bound.
>
> - **Computational Efficiency and Practical Implementation**
>
> The main computational obstacle lies in line 5 of Algorithm 1. **To implement the algorithm in practice, gradient ascent can be applied here to solve the optimization problem**. More specifically, assume that $\pi ^ 0$ and $\pi ^ 1$ are parameterized by $\nu ^ 0$ and $\nu ^ 1$ (for example, they might be parametrized by a neural network). Then the gradient of the objective $f(\nu ^ 0,\nu ^ 1) := \Vert \widehat{\phi}(\pi ^ 0) - \widehat{\phi}(\pi ^ 1) \Vert _ {{\widehat{\Sigma}} ^ {-1} _ n}$ with respect to $\nu ^ 0$ (gradient of $\nu ^ 1$ can be similarly computed) can be computed as:
>
> $$\frac{\partial f(\nu ^ 0 , \nu ^ 1)}{\partial \nu ^ 0} = 2 \frac{\partial \widehat{\phi}(\pi ^ 0)}{\partial \nu ^ 0} \cdot {\widehat{\Sigma}} ^ {-1} _ n \cdot (\widehat{\phi}(\pi ^ 0) - \widehat{\phi}(\pi ^ 1)).$$
>
> Here $\widehat{\phi} (\pi ^ 0)$ and $\widehat{\phi}(\pi ^ 1)$ can be estimated efficiently by simulating $\pi ^0 $ and $\pi ^ 1$ in $\widehat{P}$. For $\frac{\partial \widehat{\phi}(\pi ^ 0)}{\partial \nu ^ 0}$, we show in Section 4 that each coordinate of $\widehat{\phi}(\pi^0)$ is indeed a value function with respect to a certain reward function under $\widehat{P}$. Therefore we can apply the techniques in policy gradient literature such as REINFORCE to estimate $\frac{\partial \widehat{\phi}(\pi ^ 0)}{\partial \nu ^ 0}$ efficiently. We will incorporate this explanation in our paper.
>
> - **Intuition about Reward Free vs Reward Dependent Collection**
>
> We agree the phenomenon you mentioned has indeed been shown in the literature of standard reward-free exploration RL [3-4], where the sample complexity of the reward-free algorithms is only slightly worse than reward-aware algorithms [5-6]. **However, in PbML, we won’t necessarily see this phenomenon**.
>
>
> 1. **First, reward learning in PbRL is much more difficult than reward learning in standard RL** in the sense that rewards in standard RL could be just learned from regression over $r$ onto a pair of $(s,a)$ at single-time point; but, rewards in PbRL need to be learned from exploratory data over **the whole trajectories** (not a pair of $(s,a)$).  Due to the difficulty of reward learning, it is unclear whether a reward-aware adaptive data collection process leads to better sample complexity in PbRL **because learned awards could be very inaccurate before exploring the whole trajectory very well**. That’s why our algorithm has a better sample complexity than [1], even in terms of not only the sample complexity of human feedback $N_{hum}$ but also the sample complexity of transitions $N_{tra}$, as we mention at the end of Section 3.
>
>
> 2. **Second, compared to standard RL, we need to make a distinction between sample complexity in terms of human feedback $N_{hum}$ and the sample complexity of trajectories $N_{tra}$**. If we implement adaptive experimental designs depending on learned rewards, this means we need to query human feedback (to accurately learn rewards) while we explore over trajectories. This results in larger sample complexity in terms of $N_{hum}$ while we might be able to successfully decrease $N_{tra}$, which may be undesirable because in practice human feedback is often more expensive than trajectories.

---

> ### Author Response · Authors · 2023-11-16
> **Response (Part 2)**
>
> - **Insights for Regret Minimization**
>
> We think this generalization is possible. For example, we can collect trajectories for reward learning and dynamics learning separately in different frequencies. When the dynamics is more complicated, we can query human feedback and update the reward model **less often** to wait for the learning of the dynamics. This can potentially reduce the human feedback complexity.
>
> [1] Saha, Aadirupa, Aldo Pacchiano, and Jonathan Lee. "Dueling RL: Reinforcement Learning with Trajectory Preferences." International Conference on Artificial Intelligence and Statistics. PMLR, 2023.
>
> [2] Chen, Xiaoyu, et al. "Human-in-the-loop: Provably efficient preference-based reinforcement learning with general function approximation." International Conference on Machine Learning. PMLR, 2022.
>
>
> [3] Jin, Chi, et al. "Reward-free exploration for reinforcement learning." International Conference on Machine Learning. PMLR, 2020.
>
> [4] Wang, Ruosong, et al. "On reward-free reinforcement learning with linear function approximation." Advances in neural information processing systems 33 (2020): 17816-17826.
>
> [5] Azar, Mohammad Gheshlaghi, Ian Osband, and Rémi Munos. "Minimax regret bounds for reinforcement learning." International Conference on Machine Learning. PMLR, 2017.
>
> [6] Jin, Chi, et al. "Provably efficient reinforcement learning with linear function approximation." Conference on Learning Theory. PMLR, 2020.

---

> > ### Comment · Reviewer_m5cB · 2023-11-17
> > **Thank you for your response and clarifications**
> >
> > Thank you for responding to my comments and questions. I have no more questions and keep my original assessment of this paper making some interesting contributions ($(\varepsilon, \delta)$-PAC PbRL and separate trajectory selection), which are in my opinion worthy of acceptance.

---

> > > ### Author Response · Authors · 2023-11-20
> > >
> > > You are welcome, and thank you again for the positive feedback!

---

### Official Review · Reviewer_s63Y · 2023-11-01

**Soundness:** 4 excellent
**Presentation:** 3 good
**Contribution:** 2 fair
**Rating:** 6
**Confidence:** 3

**Summary:**

This paper studies online reward-agnostic Preference-based Reinforcement Learning (PbRL) with linear reward functions, which aims at learning a near-optimal policy with human preferences as feedback. This setting is different from current literature of online PbRL in that the humans are able to get rid of online preferences labeling in the exploration on-the-fly, instead they label the exploratory trajectories after they are complete collected. The paper provides sample efficient algorithms with different reward-agnostic setup: when a model-based reward-free exploration oracle is given, or when the underlying model is linear MDP or low-rank MDP. The theoretical algorithms are complemented with polynomial sample complexity in terms of the MDP parameters and $\epsilon$. Moreover, the sample complexity can be reduce to $O(\exp(B_{\mathrm{adv}}))$ where $B_{\mathrm{adv}}$ is the $l_\infty$ norm of the optimal advantage function when action-wise preference in terms of the optimal Q function is provided instead of trajectory-wise preference.

**Strengths:**

PbRL and RLHF are highly related to important practical problems such as tuning large models, and they are also proved to be one of the key designs in the success of LLMs. Therefore, the importance of studying the theoretical information provided by human preference data is significant.

**Weaknesses:**

1. Since the motivation of this work derives from prominent practical problems, the results seem to be limited to guide the use of PbRL of RLHF in real-world problems. Although the results show a reduction of the number of human preference data compared to previous work by collecting a exploratory dataset in advance, it is doubtful whether the algorithmic designs are computational efficient to be implemented to improve the RLHF. Some experiment results (even on toy examples) are appreciated to show the effectiveness of decoupling the human labeling process and online exploration stage. The significance of the results cannot be fully verified without the proof of practical usage of these algorithmic designs, because the proposed algorithms may be computational inefficient, thus explaining the theoretical advantage of such algorithms rarely helps to explain the advantage of current PbRL used in practice.

2. The setting looks like a combination of reward-free exploration and online PbRL, which are both standard in reinforcement learning and extensively studied. Therefore, it is hard to evaluate the technical contributions as a theoretical work. The subroutine to collect exploratory data with linear reward functions (using whether reward-free exploration oracle or linear reward-free exploration subroutine) used for human preferences, the MLE estimation of the underlying reward model, and the planning of the optimal policy seem to be slight modification of standard algorithms in the literature, as long as the proving tools. I recommend to explicitly explain the technical contributions in the paper.

**Questions:**

The previous work Pacchiano et al. studied the online PbRL with a linear reward function. It seems that the sample complexity of this in the reward-free exploratory stage is better than that of the previous work, which is not reward-free setting. What causes this gap between current work and previous work?

---

> ### Author Response · Authors · 2023-11-16
> **Response (Part 1)**
>
> Thank you for the valuable feedback! Here are our responses.
>
> - **Practical Insights and Computational Efficiency**
>
> 1. We first want to clarify that this work mainly focuses on the **statistical complexity** of PbRL and attempts to explain the advantages of the current practical PbRL framework from the perspective of sample complexity. For practical insights, we agree that empirical experiments can further verify our claims. However, **we do not think theoretical analysis cannot provide insights to explain the advantages of current PbRL in practice**. Instead, we believe that by studying the theoretical properties of some simplified cases (such as the linear reward parametrization setting in this paper), we are able to dig out the mathematical explanation behind the phenomenon in practice, and make actual algorithms more sample efficient.  As a matter of fact, from the sample complexity comparison against existing works, our paper provides **the following insights into practical algorithm design**: 1) **decoupling the interaction with the environment and the collection of human feedback could lead to reduced sample complexity in theory** 2) if we want to use fewer human feedback, we need to **make the query dataset as diverse as possible such that it can cover the feature space**.
>
> We do not have empirical evidence, but we would like to point out that **a lot of important theoretical works also focus on theoretical analysis and lack empirical evaluation**, from standard RL [1-5] to PbRL [6-8]. Among them, the algorithms in [3-8] are not computationally efficient either. In particular, as far as we know, [6-7] are the only existing online PbRL algorithms with provable guarantees, and their algorithms are **even harder to implement than ours in practice** because the optimization problems in their algorithms have more constraints and are much more complicated.
>
> 2. **Second, for computational efficiency, the majority of computational cost lies in line 5 in Algorithm 1. To implement the algorithm in practice, gradient ascent can be applied here to solve the optimization problem**. More specifically, assume that $\pi ^ 0$ and $\pi ^ 1$ are parameterized by $\nu ^ 0$ and $\nu ^ 1$ (for example, they might be parametrized by a neural network). Then the gradient of the objective $f(\nu ^ 0,\nu ^ 1) := \Vert \widehat{\phi}(\pi ^ 0) - \widehat{\phi}(\pi ^ 1) \Vert _ {{\widehat{\Sigma}} ^ {-1} _ n}$ with respect to $\nu ^ 0$ (gradient of $\nu ^ 1$ can be similarly computed) can be computed as:
>
> \begin{align*}
> \frac{\partial f(\nu ^ 0 , \nu ^ 1)}{\partial \nu ^ 0} = 2 \frac{\partial \widehat{\phi}(\pi ^ 0)}{\partial \nu ^ 0} \cdot {\widehat{\Sigma}} ^ {-1} _ n \cdot (\widehat{\phi}(\pi ^ 0) - \widehat{\phi}(\pi ^ 1))
> \end{align*}
>
> Here $\widehat{\phi} (\pi ^ 0)$ and $\widehat{\phi}(\pi ^ 1)$ can be estimated efficiently by simulating $\pi ^0 $ and $\pi ^ 1$ in $\widehat{P}$. For $\frac{\partial \widehat{\phi}(\pi ^ 0)}{\partial \nu ^ 0}$, we show in Section 4 that each coordinate of $\widehat{\phi}(\pi^0)$ is indeed a value function with respect to a certain reward function under $\widehat{P}$. Therefore we can apply the techniques in policy gradient literature such as REINFORCE to estimate $\frac{\partial \widehat{\phi}(\pi ^ 0)}{\partial \nu ^ 0}$ efficiently. We will incorporate this explanation in our paper.
>
> 3. **Third, although our analysis posits linear models, we can potentially extend our algorithms to the one with deep neural networks**. This is roughly done after learning representations with deep neural networks and treating it as a feature $\phi$.  This type of extension is commonly used in the literature of standard RL [9,10]. Since the focus in our paper is to establish theoretically provably efficient algorithms without any heuristics, we explain this possible extension in the section of future works.

---

> ### Author Response · Authors · 2023-11-16
> **Response (Part 2)**
>
> - **Technical Contributions**
>
> Most importantly, the novelty of our algorithm mainly lies in the **reward-agnostic trajectory collection** procedure (line 4-7 in Algorithm 1). For the other steps in the framework, we follow the common practice in empirical works and thus they are the same as the practical work. Our reward-agnostic trajectory collection procedure takes inspiration from the existing works in reward-free exploration RL and online PbRL. However, it is **not a simple combination of the existing algorithms**. More specifically, our analysis has the following challenges that are not solved in the literature. We will incorporate this explanation in the final version.
>
> 1. **First, almost all the reward-free exploration literature is focused on learning the transition dynamics of the MDPs. Our problem is completely different because our main task is to learn the reward function from human preferences**. Consequently, our iterative collection procedure (line 4-7 in Algorithm 1) also differs from the reward-free exploration literature. For example, [2] is the closest work to ours and studies the reward-free exploration in linear MDPs. They utilize a variant of UCB-LSVI, which is quite different from our collection procedure. Correspondingly, we can not directly use the existing analysis either and need to come up with new proofs.
>
> 2. **Second, our framework is also different from the existing online PbRL [6-7]**. In their algorithms, the reward-learning process and exploration phase are not decoupled and the analysis heavily relies on the utilization of **optimism**. In contrast, our algorithm **does not use optimism** and thus their analysis techniques also do not apply here.
>
> 3. Third, our analysis of  REGIME-lin differs from the one in the literature on linear MDPs. This is because while the style of our algorithm is akin to **multi-policy evaluation** in linear MDPs, the current literature of linear MDPs mainly focuses on **Q-learning-style algorithms for policy optimization** [1,2].
>
> More specifically, in REGIME-lin, we want to **evaluate all the policies in a policy class under a certain reward function**. This brings a new challenge that does not occur in [1,2]: our learning difficulty will scale with the size of the policy class $\Pi$. This forces us to find and analyze **an appropriate policy class that is able to approximate the optimal policy while retaining a relatively small size**. In addition, a key component of our analysis is the **new concentration inequality** in Lemma 13, which also differs from the existing concentration lemma in [1,2] and may be of independent interest.
>
>
> - **Reasons of the Sample Complexity Improvement**
>
> As pointed out in the paper, we believe this is because [6] **couples the reward learning process with dynamics learning**. Consequently, part of the human feedback is also utilized to learn the dynamics and thus wasted. In contrast, our algorithm **decouples these two learning processes** and human feedback is only utilized to learn the reward, leading to more efficient human feedback complexity.
>
>
>
> [1] Jin, Chi, et al. "Provably efficient reinforcement learning with linear function approximation." Conference on Learning Theory. PMLR, 2020.
>
> [2] Wang, Ruosong, et al. "On reward-free reinforcement learning with linear function approximation." Advances in neural information processing systems 33 (2020): 17816-17826.
>
> [3] Jin, C., Liu, Q., and Miryoosefi, S. (2021a). Bellman eluder dimension: New rich classes of RL problems, and sample-efficient algorithms.
>
> [4] Zanette, A., Lazaric, A., Kochenderfer, M., & Brunskill, E. (2020, November). Learning near optimal policies with low inherent bellman error. In International Conference on Machine Learning (pp. 10978-10989). PMLR.
>
> [5] Bilinear classes: A structural framework for provable generalization in rl. In International Conference on Machine Learning (pp. 2826-2836). PMLR.
>
> [6] Saha, Aadirupa, Aldo Pacchiano, and Jonathan Lee. "Dueling RL: Reinforcement Learning with Trajectory Preferences." International Conference on Artificial Intelligence and Statistics. PMLR, 2023.
>
> [7] Chen, Xiaoyu, et al. "Human-in-the-loop: Provably efficient preference-based reinforcement learning with general function approximation." International Conference on Machine Learning. PMLR, 2022.
>
> [8] Zhu, Banghua, Jiantao Jiao, and Michael I. Jordan. "Principled Reinforcement Learning with Human Feedback from Pairwise or $ K $-wise Comparisons." arXiv preprint arXiv:2301.11270 (2023).
>
> [9] Qiu, Shuang, et al. "Contrastive ucb: Provably efficient contrastive self-supervised learning in online reinforcement learning." International Conference on Machine Learning. PMLR, 2022.
>
> [10] Zhang, Tianjun, et al. "Making linear mdps practical via contrastive representation learning." International Conference on Machine Learning. PMLR, 2022.

---

> ### Author Response · Authors · 2023-11-20
>
> Dear reviewer,
>
> There are only three days left for the discussion period. Are you satisfied with our response, or do you have any other questions that we can help with?

---

> ### Author Response · Authors · 2023-11-22
>
> Dear reviewer,
>
> This is the last day of the discussion period. Are you satisfied with our response? If so, would you consider raising the score?

---

> > ### Comment · Reviewer_s63Y · 2023-11-22
> >
> > Sorry for the delay and thank you for addressing my concerns. I agree that the theoretical insights of decoupling reward learning and exploration is invaluable, and there are a number of empirical works providing strong envidence for it. I would like to raise my score accordingly. For the technical novelty part, I still believe it is natural to decide "explorative" policies given the linear parametrization of returns and preference feedback of humans by checking policies whose features are "distant" under current dataset. However, I do agree the algorithmic design of Line 4-7 in Algorithm 1 is novel beyond [2], which uses a real reward function to define the optimal policy instead of preference data.

---

> > > ### Author Response · Authors · 2023-11-22
> > >
> > > Thank you very much for the feedback! We appreciate it a lot.

---

### Meta-Review · Area_Chair_riyT · 2023-12-06

**Metareview:**

The paper studies a novel theoretical framework for preference-based RL in a reward-agnostic setting. The proposed framework first selects a batch of exploratory trajectories and then elicits human preferences for trajectories in this batch. Theoretical characterization and sample complexity bounds for different algorithms in this framework are provided. The reviewers agreed that the paper has high novelty and that the results would be of broad interest to the community. However, the reviewers also raised several concerns and questions in their initial reviews. We want to thank the authors for their responses and active engagement during the discussion phase. The reviewers appreciated the responses, which helped in answering their key questions. The reviewers have an overall positive assessment of the paper, and there is a consensus for acceptance. The reviewers have provided detailed feedback, and we strongly encourage the authors to incorporate this feedback when preparing the final version of the paper.

**Justification For Why Not Higher Score:**

The paper could possibly be given a score of "Accept (oral)" based on score calibration with other accepted papers.

**Justification For Why Not Lower Score:**

"Accept (spotlight)" is justified as the reviewers agreed that the paper has high novelty and the results would be of broad interest to the community.

---

### Decision · Program_Chairs · 2024-01-16

Accept (spotlight)